# Generalized Decision Transformer for Offline Hindsight Information Matching

**Hiroki Furuta**
The University of Tokyo
furuta@weblab.t.u-tokyo.ac.jp

**Yutaka Matsuo**
The University of Tokyo

**Shixiang Shane Gu**
Google Research

## ABSTRACT

How to extract as much learning signal from each trajectory data has been a key problem in reinforcement learning (RL), where sample inefficiency has posed serious challenges for practical applications. Recent works have shown that using expressive policy function approximators and conditioning on future trajectory information – such as future states in hindsight experience replay (HER) or returns-to-go in Decision Transformer (DT) – enables efficient learning of multi-task policies, where at times online RL is fully replaced by offline behavioral cloning (BC), e.g. sequence modeling. We demonstrate that all these approaches are doing hindsight information matching (HIM) – training policies that can output the rest of trajectory that matches some *statistics* of future state information. We present Generalized Decision Transformer (GDT) for solving any HIM problem, and show how different choices for the feature function and the anti-causal aggregator not only recover DT as a special case, but also lead to novel Categorical DT (CDT) and Bi-directional DT (BDT) for matching different statistics of the future. For evaluating CDT and BDT, we define offline multi-task state-marginal matching (SMM) and imitation learning (IL) as two generic HIM problems, propose a Wasserstein distance loss as a metric for both, and empirically study them on MuJoCo continuous control benchmarks. Categorical DT, which simply replaces anti-causal summation with anti-causal binning in DT, enables arguably the first effective offline multi-task SMM algorithm that generalizes well to unseen (and even synthetic) multi-modal reward or state-feature distributions. Bi-directional DT, which uses an anti-causal second transformer as the aggregator, can learn to model any statistics of the future and outperforms DT variants in offline multi-task IL, i.e. one-shot IL. Our generalized formulations from HIM and GDT greatly expand the role of powerful sequence modeling architectures in modern RL.

## 1 INTRODUCTION

Reinforcement learning (RL) suffers from the problem of sample inefficiency, and a central question is how to extract as much learning signals, or constraint equations (Pong et al., 2018; Tu & Recht, 2019; Dean et al., 2020), from each trajectory data as possible. As dynamics transitions and Bellman equation provide a rich source of supervisory objectives and constraints, many algorithms combined model-free with model-based, and policy-based with value-based in order to achieve maximal sample efficiency, while approximately preserving stable, unbiased policy learning (Heess et al., 2015; Gu et al., 2016; 2017; Buckman et al., 2018; Pong et al., 2018; Tu & Recht, 2019).

Orthogonal to these, in the recent years we have seen a number of algorithms that are derived from different motivations and frameworks, but share the following common trait: they use **future trajectory information** $\tau_{\mathbf{t:T}}$ to accelerate optimization of a **contextual policy** $\pi(\mathbf{a_t}|\mathbf{s_t}, \mathbf{z})$ with **context z** with respect to a **parameterized reward function** $\mathbf{r}(\mathbf{s_t}, \mathbf{a_t}, \mathbf{z})$ (see Section 3 for notations). These *hindsight* algorithms have enabled Q-learning with sparse rewards (Andrychowicz et al., 2017), temporally-extended model-based RL with Q-function (Pong et al., 2018), mastery of 6-DoF object manipulation in cluttered scenes from human play (Lynch et al., 2019), efficient multi-task RL (Eysenbach et al., 2020; Li et al., 2020), offline self-supervised discovery of manipulation primitives from pixels (Chebotar et al., 2021), and offline RL using return-conditioned supervised

learning with transformers (Chen et al., 2021a; Janner et al., 2021). We derive a generic problem formulation covering all these variants, and observe that this **hindsight information matching (HIM)** framework, *with behavioral cloning (BC) as the learning objective*, can learn a conditional policy to generate trajectories that each satisfy any properties, including distributional.

Given this insight and recent casting of RL as sequence modeling (Chen et al., 2021a; Janner et al., 2021), we propose Generalized Decision Transformer (GDT), a family of algorithms for future information matching using **hindsight behavioral cloning with transformers**, and greatly expand the applicability of transformers and other powerful sequential modeling architectures within RL **with only small architectural changes to DT**. In summary, our key contributions are:

- We introduce hindsight information matching (HIM) (Section 4, Table 1) as a unifying view of existing hindsight-inspired algorithms, and Generalized Decision Transformers (GDT) as a generalization of DT for RL as sequence modeling to solve *any* HIM problem (Figure 1).
- Inspired by distribution RL (Bellemare et al., 2017; Dabney et al., 2018) and state-marginal matching (SMM) (Lee et al., 2020; Ghasemipour et al., 2020; Gu et al., 2021), we define *offline multi-task SMM* problems, propose Categorical DT (CDT) (Section 5), validate its empirical performance to match feature distributions (even generalizing to a synthetic bi-modal target distribution at times), and construct the first benchmark tasks for offline multi-task SMM.
- Inspired by one-shot imitation learning (Duan et al., 2017; Finn et al., 2017; Dasari & Gupta, 2020), we define *offline multi-task imitation learning (IL)*, propose a Wasserstein-distance evaluation metric, develop Bi-directional DT (BDT) as a fully expressive variant of GDT (Section 5), and demonstrate BDT's competitive performance at offline multi-task IL.

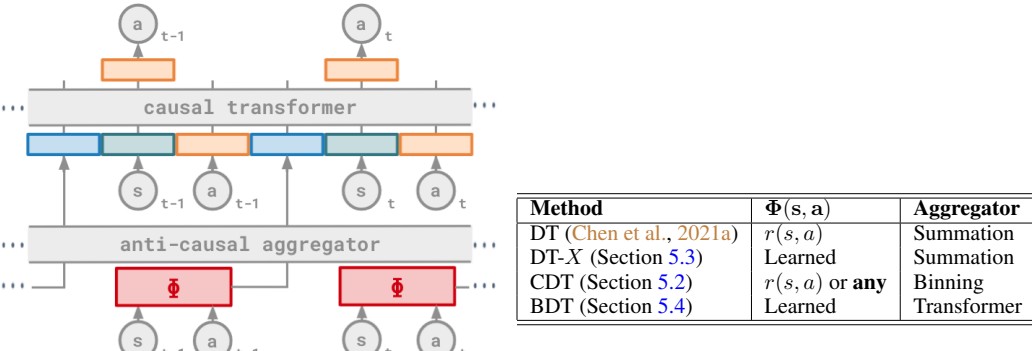

| Method | $\Phi(\mathbf{s}, \mathbf{a})$ | Aggregator |
|---|---|---|
| DT (Chen et al., 2021a) | $r(s, a)$ | Summation |
| DT-*X* (Section 5.3) | Learned | Summation |
| CDT (Section 5.2) | $r(s, a)$ or **any** | Binning |
| BDT (Section 5.4) | Learned | Transformer |

Figure 1: Generalized Decision Transformer (GDT), where the figure is a minor generalization of the DT architecture (Chen et al., 2021a) and the table summarizes how it leads to different classes of algorithms **with only small architectural changes**. If the feature function $\Phi(s, a)$ is reward $r(s, a)$ and the anti-causal aggregator is $\gamma$-discounted summation, we recover DT for offline RL. If the aggregator is binning, we get Categorical DT (CDT) for offline multi-task state-marginal matching. If the aggregator is a second transformer, we get Bi-directional DT (BDT) for offline multi-task imitation learning (IL), or equivalently one-shot IL. The choices of $\Phi(s, a)$ and the aggregator together decide $I^\Phi(\tau)$ in Hindsight Information Matching (HIM) objective discussed in Section 4 and Table 1, where conversely GDT can essentially solve any HIM problem with proper choices of $\Phi$ and aggregator.

## 2 RELATED WORK

**Hindsight Reinforcement Learning and Behavior Cloning** Hindsight techniques (Kaelbling, 1993; Andrychowicz et al., 2017; Pong et al., 2018) have revolutionized off-policy optimization with respect to parameterized reward functions. Two key insights were (1) for off-policy algorithms such as Q-learning (Mnih et al., 2015; Gu et al., 2016) and actor-critic methods (Lillicrap et al., 2016; Haarnoja et al., 2018; Fujimoto et al., 2018; Furuta et al., 2021a), the same transition samples can be used to learn with respect to any reward parameters, as long as the reward function is re-computable, i.e. "relabel"-able, like goal reaching rewards, and (2) if policy or Q-functions are smooth with respect to the reward parameter, generalization can speed up learning even with respect to "unexplored" rewards. In goal-based RL where future states can inform "optimal" reward parameters with respect to the transitions' actions, hindsight methods were applied successfully to

enable effective training of goal-based Q-function for sparse rewards (Andrychowicz et al., 2017), derive exact connections between Q-learning and classic model-based RL (Pong et al., 2018), data-efficient off-policy hierarchical RL (Nachum et al., 2018), multi-task RL (Eysenbach et al., 2020; Li et al., 2020), offline RL (Chebotar et al., 2021), and more (Eysenbach et al., 2021; Choi et al., 2021; Ren et al., 2019; Zhao & Tresp, 2018; Ghosh et al., 2021; Nasiriany et al., 2021). Additionally, Lynch et al. (2019) and Gupta et al. (2018) have shown that often BC is sufficient for learning generalizable parameterized policies, due to rich positive examples from future states, and most recently Chen et al. (2021a) and Janner et al. (2021), when combined with powerful transformer architectures (Vaswani et al., 2017), it produced state-of-the-art offline RL and goal-based RL results. Lastly, while motivated from alternative mathematical principles and not for parameterized objectives, future state information was also explored as ways of reducing variance or improving estimations for generic policy gradient methods (Pinto et al., 2017; Guo et al., 2021; Venuto et al., 2021).

**Distributional Reinforcement Learning and State-Marginal Matching** Modeling the full distribution of returns instead of the averages led to the development of distributional RL algorithms (Bellemare et al., 2017; Dabney et al., 2018; 2020; Castro et al., 2018; Barth-Maron et al., 2018) such as Categorical Q-learning (Bellemare et al., 2017). While our work shares techniques such as discretization and binning, these works focus on optimizing a non-conditional reward-maximizing RL policy and therefore our problem definition is closer to that of state-marginal matching algorithms (Hazan et al., 2019; Lee et al., 2020; Ghasemipour et al., 2020; Gu et al., 2021), or equivalently inverse RL algorithms (Ziebart et al., 2008; Ho & Ermon, 2016; Finn et al., 2016; Fu et al., 2018; Ghasemipour et al., 2020) whose connections to feature-expectation matching have been long discussed (Abbeel & Ng, 2004). However, those are often exclusively online algorithms even sample-efficient variants (Kostrikov et al., 2019), since density-ratio estimations with either discriminative (Ghasemipour et al., 2020) or generative (Lee et al., 2020) approach requires on-policy samples, with a rare exception of Kostrikov et al. (2020). Building on the success of DT and brute-force hindsight imitation learning, our Categorical DT is to the best our knowledge the first method that benchmarks offline state-marginal matching problem in the multi-task settings.

**RL and Imitation Learning as Sequence Modeling** When scaled to the extreme levels of data and computing, sequence models such as transformers (Vaswani et al., 2017) can train models to master an impressive range of capabilities in natural language processing and computer vision (Devlin et al., 2019; Radford et al., 2019; Brown et al., 2020; Radford et al., 2021; Ramesh et al., 2021; Chen et al., 2021b; Bommasani et al., 2021; Dosovitskiy et al., 2020). Comparing to their popularity in other areas, the adoption of transformers or architectural innovations in RL have been slow, partially due the difficulty of using transformers over temporal scales for online RL (Parisotto et al., 2020). Recent successes have focused on processing variable-length per-timestep information such as morphology (Kurin et al., 2021), sensory information (Tang & Ha, 2021), one-shot or few-shot imitation learning (Dasari & Gupta, 2020), or leveraged offline learning (Chen et al., 2021a; Janner et al., 2021). Our formulation enables sequence modeling to solve novel RL problems such as state-marginal matching with minimal architectural modifications to DT, greatly expanding the impacts of transformers and other powerful sequence models in RL.

## 3 PRELIMINARIES

We consider a Markov Decision Process (MDP) defined by the tuple of action space $\mathcal{A}$, state space $\mathcal{S}$, transition probability function $p(s'|s, a)$, initial state distribution $p(s_0)$, reward function $r(s, a)$, and discount factor $\gamma \in (0, 1]$. In deep RL, a policy that maps the state space to the action space is parameterized by the function approximators, $\pi_\theta(a|s)$[1]. The RL objective is given by:

$$L_{\text{RL}}(\pi) = \frac{1}{1-\gamma}\mathbb{E}_{s \sim \rho^\pi(s), a \sim \pi(\cdot|s)}\left[r(s, a)\right] \qquad (1)$$

where $p_t^\pi(s) = \iint_{s_{0:t}, a_{0:t-1}} \prod_t p(s_t|s_{t-1}, a_{t-1})\pi(a_t|s_t)$ and $\rho^\pi(s) = (1-\gamma)\sum_{t'} \gamma^{t'} p_{t'}^\pi(s_{t'} = s)$ are short-hands for time-aligned and time-aggregated state marginal distributions following policy $\pi$.

---

[1]For simplicity of notations, we write Markovian policies; however, such notations can easily apply to non-Markov policies such as Decision Transformer (Chen et al., 2021a) by converting to an augmented MDP consisting of past $N$ states, where $N$ is the context window of DT.

## 3.1 STATE MARGINAL MATCHING

State marginal matching (SMM) (Lee et al., 2020; Hazan et al., 2019; Ghasemipour et al., 2020) has been recently studied as an alternative problem specification in RL, where instead of stationary-reward maximization, the objective is to find a policy minimizing the divergence $D$ between its state marginal distribution $\rho^\pi(s)$ to a given target distribution $p^*(s)$[2]:

$$L_{\text{SMM}}(\pi) = -D(\rho^\pi(s), p^*(s)) \tag{2}$$

where $D$ is a divergence measure such as Kullback-Leibler (KL) divergence (Lee et al., 2020; Fu et al., 2018) or, more generally, some $f$-divergences (Ghasemipour et al., 2020). For the target distribution $p^*(s)$, Lee et al. (2020) set a uniform distribution to enhance the exploration over the entire state space; Ghasemipour et al. (2020) and Gu et al. (2021) set through scripted *distribution sketches* to generate desired behaviors; and adversarial inverse RL methods (Ho & Ermon, 2016; Fu et al., 2018; Ghasemipour et al., 2020; Kostrikov et al., 2020) set as the expert data for imitation learning. Notably, unlike the RL objective in Eq.1, SMM objectives like Eq.2 no longer depend on task rewards and are only functions of state transition dynamics and target state distribution.

## 3.2 PARAMETERIZED RL OBJECTIVES

Lastly, we discuss the basis for methods like HER and TDM (Andrychowicz et al., 2017; Pong et al., 2018), LfP (Lynch et al., 2019), and return-conditioned or upside-down RL (Srivastava et al., 2019; Kumar et al., 2019; Chen et al., 2021a; Janner et al., 2021): parameterized RL objectives. Given parameterized reward functions with parameter $z \in \mathcal{Z}$, a conditional policy $\pi(a|s, z)$ is learned with respect to multiple values of $z$ simultaneously weighted by $p(z)$. As examples, the RL objective in Eq.1 becomes:

$$L_{\text{RL}}(\pi) = \mathbb{E}_z \left[ L_{\text{RL}}(\pi, z) \right] = \frac{1}{1 - \gamma} \mathbb{E}_{z \sim p(z), s \sim \rho_z^\pi(s), a \sim \pi(\cdot|s, z)} \left[ r_z(s, a) \right] \tag{3}$$

where the state marginal $\rho_z^\pi$ is from rolling out a conditioned policy $\pi(\cdot|\cdot, z)$. These can be considered as a special case of contextual MDPs (Jiang et al., 2017) and are all *multi-task* RL problems.

# 4 HINDSIGHT INFORMATION MATCHING

We show how HER and TDM (Andrychowicz et al., 2017; Pong et al., 2018), LfP (Lynch et al., 2019), hindsight multi-task RL (Li et al., 2020; Eysenbach et al., 2020), and return-conditioned or upside-down RL (Srivastava et al., 2019; Kumar et al., 2019; Chen et al., 2021a; Janner et al., 2021) all belong to *hindsight* algorithms with a shared idea of **using future state information to automatically mine for positive, or "optimal", examples with respect to certain contextual parameter values**, where these examples can accelerate RL or be used for behavior cloning (BC), i.e. supervised learning. We start by defining additional notations.

Given a partial trajectory from state $s_t$ as $\tau_t = \{s_t, a_t, s_{t+1}, a_{t+1}, \dots\}$, we define its *information statistics* as $I(\tau_t)$. $I(\tau_t)$ **could be any function of a trajectory** that captures some statistical properties in state-space or trajectory-space, such as sufficient statistics of a distribution, like mean, variance or higher-order moments (Wainwright & Jordan, 2008). For convenience, we further define the notion of a feature function $\Phi(\cdot, \cdot) : S \times A \to F$[5], where the trajectory is then noted as $\tau_t^\Phi = \{\phi_t, \phi_{t+1}, \dots, \phi_T\}, \phi_t = \Phi(s_t, a_t) \in F$ and the information statistics as $I^\Phi(\tau_t)$. $\Phi$ in practice

---

[2]As discussed in Fu et al. (2018) and Ghasemipour et al. (2020), it's also straight-forward to define state-action-marginal matching with respect to $\rho^\pi(s, a) = \rho^\pi(s)\pi(a|s)$ and the exact same algorithms apply.

[3]There is a rich literature on one-shot, or few-shot, imitation learning through meta learning. For closer connections to parameterized policies and relabeling, we mainly discuss metric-based (or amortization-based) methods (Duan et al., 2016), as opposed to gradient-based approaches (Finn et al., 2017).

[4]DisCo RL (Nasiriany et al., 2021) conditions on a parameterized goal distribution and uses hindsight techniques; however, their RL objective in Equation 1 $\mathbb{E}_{s \sim \rho_z^\pi(s)} \left[ \log p_z^*(s) \right]$, in contrast to a proper divergence objective like in Ghasemipour et al. (2020), is missing the $\mathcal{H}\left(\rho_z^\pi(s)\right)$ entropy term and is essentially just solving for parameterized stationary reward maximization, as also stated in their Remark 1.

[5]Recent mutual information maximization or empowerment methods (Eysenbach et al., 2019; Sharma et al., 2020; Choi et al., 2021) also make similar assumptions; see Gu et al. (2021) for more details.

| Method | Algo. Type | Training | $\mathbf{I}^{\Phi}(\tau)$ | Architectures |
|---|---|---|---|---|
| Andrychowicz et al. (2017) | RL | Online | $\phi_T$ | MLP |
| Pong et al. (2018) | RL | Online | $\phi_T$ | MLP |
| Chebotar et al. (2021) | RL | Offline | $\phi_T$ | CNN |
| Li et al. (2020) | RL | Online | $\arg\max \sum_t \gamma^t r(s_t, a_t, \cdot)$ | MLP |
| Eysenbach et al. (2020) | BC/RL | On/Offline | $\arg\max \sum_t \gamma^t r(s_t, a_t, \cdot)$ | MLP |
| Lynch et al. (2019) | BC | Offline | $\phi_T$ | Stochastic RNN |
| Ghosh et al. (2021) | BC | Online | $\phi_T$ | MLP |
| Srivastava et al. (2019) | BC | Online | $\sum_t \gamma^t r_t$ | Fast Weights |
| Kumar et al. (2019) | BC | Online | $\sum_t \gamma^t r_t$ | MLP |
| Janner et al. (2021) | BC | Offline | $\sum_t \gamma^t r_t$ or $\phi_T$ | Transformer |
| Duan et al. (2017)[3] | BC | Offline | $\tau$ | MLP + LSTM |
| Generalized DT (ours) | BC | Offline | Any | Transformer |
| DT (Chen et al., 2021a) | BC | Offline | $\sum_t \gamma^t r_t$ | Transformer |
| Categorical DT (ours)[4] | BC | Offline | $\text{histogram}(r_t, \gamma)$ | Transformer |
| Bi-Directional DT (ours) | BC | Offline | $\tau$ | Transformer |

Table 1: A coarse summary of hindsight information matching (HIM) algorithms. The notation follows Section 4. With HIM, all prior works can be categorized to four generic problem types based on $I^{\Phi}(\tau)$: (1) **goal-based** $\phi_T$ (Andrychowicz et al., 2017), (2) **multi-task** $\arg\max \sum_t \gamma^t r(s_t, a_t, \cdot)$ (Li et al., 2020), (3) **return-based** $\sum_t \gamma^t r_t$ (Chen et al., 2021a), or (4) **full trajectory imitation** $\tau$ (Duan et al., 2017). $\Phi$ is the reward function $r(s, a)$ in (2) and (3), an indexing function for state dimensions (e.g. xy-velocities) or a learned function (Nair et al., 2018) in (1), or an identify function in (4). Our CDT introduces a new category, (5) **distribution-based** $I^{\Phi}(\tau) = \text{histogram}(r_t, \gamma)$, based on a minimal modification to DT, while our BDT can be considered as DT adapted for (4), the trajectory imitation.

can be an identity function, the reward function $r(s, a)$, sub-dimensions of $s$ (e.g. xy-velocities), or a generic parameterized function (e.g. auto-encoder). Generalizing reward-centric intuitions in DT (Chen et al., 2021a), we define *information matching (IM)* problems as learning a conditional policy $\pi(a|s, z)$ whose trajectory rollouts satisfy some desired information statistics value $z$:

$$\min_\pi \mathbb{E}_{z \sim p(z), \tau \sim \rho_z^\pi(\tau)} \left[ D(I^{\Phi}(\tau), z) \right] \tag{4}$$

An important observation for the IM objective (Eq.4) is that **for any given trajectory $\tau$, setting $z^* = I^{\Phi}(\tau)$ will minimize the inner term divergence $D = 0$ and therefore $\tau$ states and actions are optimal with respect to $z = z^*$ and samples of $(\tau_i, z_i^*)$ can be used to accelerate RL or do BC**. We call these algorithms *hindsight information matching (HIM)* algorithms.

Table 1, which classifies prior methods into effectively four categories based on $\mathbf{I}^{\Phi}(\tau)$, leads us to have the following insights around HIM algorithms:

- New HIM algorithms can be proposed by simply changing $\mathbf{I}^{\Phi}(\tau)$, as we did to propose Categorical DT for (5) distribution-based.
- Given a choice of $\mathbf{I}^{\Phi}(\tau)$, new HIM algorithms can be proposed by changing implementation details (Furuta et al., 2021a), such as using "RL" or "BC" as **algorithm type**, doing "Online" or "Offline" **training** (Levine et al., 2020), and network **architectures**. All "Offline" "BC" methods could be adopted easily to "Online" learning through recursive data collections (Ghosh et al., 2021; Kumar et al., 2019; Matsushima et al., 2021).
- Only (1) goal-based and (2) multi-task can use "RL" as **algorithm type**, while all four, plus our (5) distribution-based, can use "BC", because "RL" requires optimizing Eq. 4 with respect to the policy, which gets non-trivial for some choices of $\mathbf{I}^{\Phi}(\tau)$; e.g. (3-5) return-based, full trajectory imitation, or distribution-based. "BC" bypasses the need to solve Eq. 4 and therefore is universally applicable to any $\mathbf{I}^{\Phi}(\tau)$ or HIM algorithm[6].

## 5 GENERALIZED DECISION TRANSFORMER

Following the insights in Section 4, we introduce **Generalized Decision Transformer (GDT)**, which generalizes DT (Chen et al., 2021a) based on different choices of $I^{\Phi}(\tau)$, as described in Figure 1 and the last rows of Table 1. We chose DT as the base model since it is a simple model that uses "BC" as the algorithm type and "Transformer" as the architecture. The choice of "BC" is a must, so we can tractably train GDT with respect to any $I^{\Phi}(\tau)$ or HIM problem. The choice for the architecture is

---

[6]Evolutionary strategies (Salimans et al., 2017), technically a non-RL black-box algorithm, could be applied, but accurate estimations of $D$ in Eq.4 could require prohibitively many samples.

more flexible; however, we decided to use transformers (Vaswani et al., 2017) in this work due to their enormous scaling successes in language and vision domains (Dosovitskiy et al., 2020; Brown et al., 2020; Ramesh et al., 2021). See Algorithm 1 (in Appendix F) for the full pseudocode. While GDT in Figure 1 can lead to different algorithms depending on different choices of the feature function $\Phi(s, a)$ and the anti-causal aggregator (which together determine $I^{\Phi}(\tau)$), in this work we focus our empirical studies on the following two variants: Categorical DT (CDT) and Bi-directional DT (BDT).

## 5.1 Task Definitions and Metrics

Before proceeding to define CDT and BDT, we first concretely define the tasks they are designed to solve, namely: **offline multi-task state-marginal matching (SMM)**, and **offline multi-task imitation learning (IL)**. Given the intrinsic connection or equivalence between distribution matching and IL (Ghasemipour et al., 2020), these two separate terminologies may seem redundant. However, inspired by the initial papers studying SMM problems (Lee et al., 2020; Ghasemipour et al., 2020) which qualitatively evaluates distribution matching results in specified state dimensions (e.g. xy-positions), we define the imitation task as offline multi-task SMM if specific $\Phi$ is given, and as offline multi-task IL if $\Phi$ is an identity (i.e. $\phi = s$) or learned (e.g. auto-encoder). We essentially view IL as SMM evaluation on full state.

Given these definition, we also define a single metric for both offline multi-task SMM/IL: typical IL assumes some availability of task reward or success evaluation, and indirectly measure the quality of imitation through it (Ho & Ermon, 2016; Fu et al., 2018). Instead, again grounding on its connection to distribution matching (Ghasemipour et al., 2020), we propose a Wasserstein loss between state-marginal and target distributions as SMM-inspired metrics for evaluating offline multi-task SMM or IL tasks. However, it is often intractable to measure such loss for full state or even for some state dimensions analytically because both state-marginal and target distributions can be non-parametric and we cannot access their densities. In practice, we empirically estimate it employing the binning of the feature space we specified. More discussions are included in Appendix C.

## 5.2 Categorical Decision Transformer for Distribution Matching

Inspired by the recent successes in distributional RL (Bellemare et al., 2017; Dabney et al., 2018; 2020), offline RL (Fujimoto et al., 2019; Jaques et al., 2020; Ghasemipour et al., 2021; Fujimoto & Gu, 2021) and state-marginal matching (Lee et al., 2020; Ghasemipour et al., 2020; Gu et al., 2021), we introduce Categorical DT (CDT) for offline state-marginal matching (SMM) problem in Section 5.1. Following the prior works (Bellemare et al., 2017; Furuta et al., 2021b), we assume low-dimensional $\Phi$, e.g. rewards or state dimensions like xyz-velocities, and employ the discretization of feature spaces to form categorical approximations of continuous distributions. To compute $z_t^* = I^{\Phi}(\tau_{t:T})$ for all timesteps $t$ given a trajectory $\tau_{1:T}$, we use similar recursive Bellman-like computation inspired by Bellemare et al. (2017) (see Appendix F for the details). To the best of our knowledge, this is the first paper to study offline multi-task SMM and propose an effective algorithm for it.

## 5.3 Decision Transformer with Learned $\Phi$

While CDT assumes some low-dimensional $\Phi$ is provided for tractable binning and distribution approximation, we also study cases where $\Phi$ or $r$ is not provided, and instead $\Phi$ is learned through auto-encoding (Hinton & Salakhutdinov, 2006; Bengio et al., 2012) (DT-AE) or contrastive (van den Oord et al., 2018; Srinivas et al., 2020; Yang & Nachum, 2021) (DT-CPC) losses for DT (see Appendix G for the details). In this case, CDT is unnecessary because if $\Phi$ learns sufficient features of $s$, matching their means, i.e. *moments*, through DT is enough to match any distribution to an arbitrary precision (Wainwright & Jordan, 2008; Li et al., 2015). Since $\Phi$ is differentiable with respect to DT's action-prediction losses, we also compare DT-E2E, where we learn $\Phi$ through end-to-end differentiation. As Section 5.1 defines, these methods do offline multi-task SMM with full state, or offline multi-task IL, a similar objective to state-marginal matching or adversarial inverse RL (Ho & Ermon, 2016; Ghasemipour et al., 2020) in online RL. To the best of our knowledge, this is the first offline multi-task IL method that explicitly accounts for SMM through architectural bottlenecks.

## 5.4 Bi-directional Decision Transformer for One-Shot Imitation Learning

The absence of given $\Phi$ could be tackled with learning not only parameterized $\Phi$ as in Section 5.3, but also a parameterized aggregator. Building on the connection to one-shot imitation learning (Duan et al., 2017), we provide another natural extension of DT under GDT framework called **Bi-directional Decision Transformer (BDT)**, which assumes an identity $\Phi$, and learns representation within the aggregator, in this case a second (anti-causal) transformer (Radford et al., 2018) that takes a reverse-order state sequence as an input. See Algorithm 1 in Appendix F for the pseudocode, and Appendix D for further comments on the connection to one-shot or meta learning. While we found some positive results for even simple unsupervised regularizer approaches (e.g. DT-AE), we observe BDT could achieve substantially better offline multi-task IL results than DT-$X$ variants in Section 5.3.

## 6 EXPERIMENTS

We empirically investigate the following questions:

- (SMM) Can CDT match unseen reward distributions?
- (SMM) Can CDT match and generalize to unseen 1D/2D state-feature distributions?
- (SMM) Can CDT match unseen synthesized state-feature distributions?
- (IL) Can BDT perform offline one-shot imitation learning in full state?

We experiment on the OpenAI Gym, MuJoCo tasks (HalfCheetah, Hopper, Walker2d, Ant-v3), a common benchmark for continuous control (Brockman et al., 2016; Todorov et al., 2012). Through the experiments, we use medium-expert datasets in D4RL (Fu et al., 2020) to ensure the decent data coverage. We sort all the trajectories by their cumulative rewards, hold out five best trajectories and five 50 percentile trajectories as a test set (10 trajectories in total), and use the rest as a train set. We report the results averaged over 20 rollouts every 4 random seed. We share our implementation to ensure the reproducibility[8].

As discussed in Section 5.1 and Appendix C, we evaluate CDT/BDT with approximate distribution matching objective: Wasserstein-1 distance between categorical distributions of features in rollouts or target trajectories. We compare CDT to DT, Meta-BC, and FOCAL (Li et al., 2021), a metric-based offline meta RL method, as baselines. While Meta-BC and FOCAL does not solve the offline distribution matching problem directly, they provide decent baseline performance since their offline one-shot adaptation to the given target trajectories could deal with it (see Appendix B for the details).

### 6.1 REWARD AND STATE-FEATURE MATCHING

First, we evaluate whether CDT could match its rollout to the target distribution. We choose reward and state-feature, such as x-velocity of the agents, as feature spaces to match. To specify the target distributions during the evaluation, we feed the categorical representation of the target to CDT. As the same as the reward case, DT takes the summation of the state-feature over a trajectory as an input.

We quantitatively compare CDT against baselines (Table 2) in x-velocity case, where CDT shows better matching results to the target distributions unseen during training. We provide the reward distribution results and the visualization in Appendix E.1, where CDT performs very well in all cases. To test the generalization additionally, we intervene the target distributions by (1) shifting the target distributions in the test set by constant offsets, and (2) synthesizing novel distributions via python scripts. See Appendix E.7 and E.8 for the results. Furthermore, to investigate the scalability of CDT to multi-dimensional state-features, we experiment 2D state-feature distribution matching (xy-velocities on Ant) in Appendix E.3, where CDT outperforms other baselines.

| Method | halfcheetah | | | hopper | | | walker2d | | | Average |
|---|---|---|---|---|---|---|---|---|---|---|
| | Expert | Medium | Total | Expert | Medium | Total | Expert | Medium | Total | |
| Categorical DT | $0.633 \pm 0.329$ | $0.996 \pm 1.467$ | $0.814 \pm 1.079$ | $0.139 \pm 0.043$ | $0.059 \pm 0.013$ | $0.099 \pm 0.051$ | $0.122 \pm 0.071$ | $0.136 \pm 0.045$ | $0.129 \pm 0.060$ | 0.347 |
| DT | $0.746 \pm 0.380$ | $1.076 \pm 1.549$ | $0.911 \pm 1.140$ | $0.177 \pm 0.053$ | $0.093 \pm 0.037$ | $0.135 \pm 0.063$ | $0.083 \pm 0.031$ | $0.146 \pm 0.084$ | $0.115 \pm 0.070$ | 0.387 |
| BC (no-context) | $3.017 \pm 0.891$ | $3.468 \pm 1.271$ | $3.242 \pm 1.121$ | $0.652 \pm 0.264$ | $0.248 \pm 0.199$ | $0.450 \pm 0.309$ | $0.748 \pm 0.529$ | $0.858 \pm 0.617$ | $0.803 \pm 0.577$ | 1.498 |
| Meta-BC | $0.852 \pm 0.688$ | $0.840 \pm 1.139$ | $0.846 \pm 0.941$ | $0.799 \pm 0.505$ | $0.130 \pm 0.056$ | $0.464 \pm 0.491$ | $0.110 \pm 0.082$ | $1.462 \pm 1.136$ | $0.786 \pm 1.052$ | 0.699 |
| FOCAL (Li et al., 2021) | $1.643 \pm 0.461$ | $1.123 \pm 1.550$ | $1.383 \pm 0.518$ | $1.456 \pm 0.473$ | $0.484 \pm 0.382$ | $0.970 \pm 0.649$ | $1.571 \pm 0.563$ | $0.603 \pm 0.427$ | $1.087 \pm 0.695$ | 1.147 |

Table 2: Quantitative evaluation of state-feature distribution matching, measuring Wasserstein-1 distance between the rollout and target distributions. We compare Categorical DT against DT, BC, Meta-BC, and FOCAL. CDT achieves better matching than baselines. See Table 9 in Appendix E.1 for the reward distribution results.

---

[8] https://github.com/frt03/generalized_dt

## 6.2 GENERALIZATION TO UNSEEN TARGET DISTRIBUTION

The performance of offline methods in RL is often restricted by the coverage or quality of datasets. While we demonstrate CDT can perform offline state-marginal matching to unseen target distributions in Section 6.1, the standard offline datasets might not be diverse enough to observe the generalization since those are collected by single-task reward-maximization policies. To test the generalization to more diverse behaviors, we investigate the following tasks: (1) z-velocity distribution matching with synthesized bi-modal behavior and (2) cheetah-velocity matching problem from meta RL/IL literature (Rakelly et al., 2019; Li et al., 2021; Fakoor et al., 2020)

### 6.2.1 SYNTHESIZING UNSEEN BI-MODAL DISTRIBUTION

To generate diverse behaviors for z-axis, we obtain the expert cheetah that backflips towards -x direction by modifying reward function (see Appendix E.4 for the details). Combining expert backflipping trajectories and expert running forward trajectories from D4RL dataset, we construct a novel dataset with diverse behaviors. We experiment the offline SMM with not only each uni-modal behavior (backflipping or running forward), but also synthesized bi-modal behavior; running forward first, then backflipping during a single rollout, using patchworked target trajectories.

Table 3 and Figure 2 (a) show that CDT successfully matches the distribution to both uni-modal (running forward or backflipping) and synthesized bi-modal distributions better than DT and FOCAL, and is comparable to Meta-BC that originally designed to deal with such multi-task settings. Due to the difficulty for RL algorithms to maximize Eq. 4, FOCAL struggles to solve the offline multi-task SMM, even though FOCAL uses the same context embedding as Meta-BC. The bi-modal behavior learned by CDT can be seen at https://sites.google.com/view/generalizeddt.

| Method | Uni-modal | Bi-modal | Average |
|---|---|---|---|
| Categorical DT | $1.562 \pm 0.632$ | $1.625 \pm 0.902$ | 1.594 |
| DT | $2.676 \pm 0.765$ | $2.703 \pm 0.703$ | 2.690 |
| Meta-BC | $1.519 \pm 0.696$ | $1.655 \pm 0.990$ | 1.587 |
| FOCAL (Li et al., 2021) | $2.203 \pm 1.050$ | $1.983 \pm 0.948$ | 2.093 |

Table 3: State-feature (z-velocity) distribution matching with uni-modal and (synthesized) bi-modal target trajectories in HalfCheetah environment. Categorical DT matches both uni- and bi-modal trajectories better than DT and FOCAL, and is comparable to Meta-BC that originally aims to solve multi-task problem.

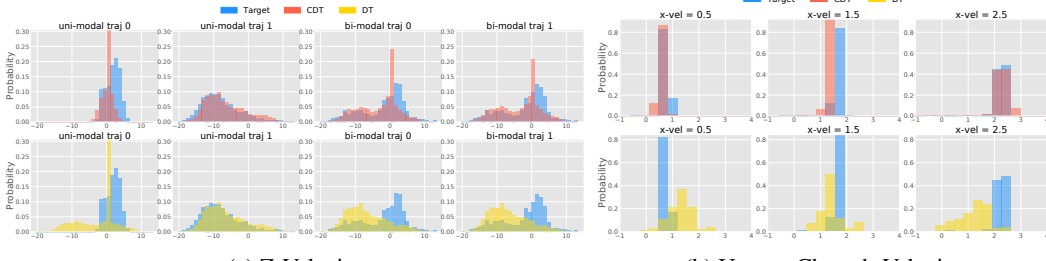

(a) Z-Velocity                    (b) Unseen Cheetah-Velocity

Figure 2: (a) Z-Velocity and (b) Unseen Cheetah-Velocity results. Blue histograms represent target distributions. In (a), CDT (red) can match not only uni-modal behaviors for both running forward and backflipping, but also bi-modal behaviors; during a single rollout running forward first, then backflipping. DT (yellow) tends to lean backflipping and fails to fit neither uni-modal nor bi-modal ones. In (b), CDT successfully handles the trajectories unseen during training, while DT seems to output covering behaviors over the dataset support.

### 6.2.2 DIVERSE UNSEEN DISTRIBUTION FROM META LEARNING TASK

Generalization to unknown target demonstrations or tasks has been actively investigated in meta or one-shot RL/IL literature (Duan et al., 2016; Wang et al., 2016). To verify the generalization of CDT to diverse behaviors, we adopt the cheetah-velocity task; a popular task in meta RL/IL (Rakelly et al., 2019; Li et al., 2021; Fakoor et al., 2020), where the cheetah tries to run with the specified velocity. We prepare 31 target x-velocities; taken from [0.0, 3.0], uniformly spaced at 0.1 intervals, and hold out $\{0.5, 1.5, 2.5\}$ as a test set. See Appendix E.5 for the dataset generation.

Table 4 and Figure 2 (b) reveal that CDT outperforms DT or FOCAL, and is slightly better than Meta-BC through the distribution matching evaluation, which implies CDT could solve offline multi-task SMM generalizing to the unknown target trajectories, given sufficiently diverse offline datasets.

| Method | x-vel: 0.5 | x-vel: 1.5 | x-vel: 2.5 | Average |
|---|---|---|---|---|
| Categorical DT | $0.060 \pm 0.026$ | $0.211 \pm 0.022$ | $0.149 \pm 0.110$ | 0.140 |
| DT | $1.197 \pm 0.227$ | $0.533 \pm 0.105$ | $0.861 \pm 0.247$ | 0.864 |
| Meta-BC | $0.150 \pm 0.069$ | $0.152 \pm 0.127$ | $0.167 \pm 0.055$ | 0.156 |
| FOCAL (Li et al., 2021) | $0.472 \pm 0.005$ | $0.952 \pm 0.073$ | $0.346 \pm 0.186$ | 0.590 |

Table 4: Generalization to unseen target velocity (x-velocity = 0.5, 1.5, 2.5) among training dataset (x-velocity $\in [0.0, 3.0] \setminus \{0.5, 1.5, 2.5\}$; uniformly spaced at 0.1 intervals), which is a popular setting in meta RL/IL. CDT successfully deals with unseen target velocities well, outperforming DT and FOCAL, and is slightly better than Meta-BC through the offline multi-task SMM evaluation.

## 6.3 ONE-SHOT DISTRIBUTION MATCHING IN FULL STATE

Lastly, we investigate BDT for offline multi-task IL, where we do not observe rewards nor state features explicitly. Instead, target full state trajectories that the agents are expected to mimic is given. We compare BDT against parameterized $\Phi$ variants; DT-AE, -CPC, and -E2E discussed in Section 5.3. We consider three strategies to train the encoder for DT-AE and -CPC: training with only unsupervised loss, training with unsupervised and DT's supervised loss jointly (called as "joint"), and pre-training with only unsupervised loss and freezing the weights during DT training (called as "frozen"). We aggregate the learned $\Phi$ by summation within the input sequence (length is 20). We evaluate BDT and DT-$X$ with the same distribution matching evaluation as Section 6.1.

We sweep $m$-dim learned feature from the encoder $\Phi(s)$ with $m \in \{1, 4, 16\}$. Table 5 presents the offline multi-task IL results with respect to x-velocity distribution, using $m = 16$ embeddings (see Appendix E.6 for the full results including $m = 1, 4$, and the reward distribution case). Similar to the discussion in Yang & Nachum (2021), DT-CPC sometimes fails to obtain sufficient representation for imitation. While even simple approaches, DT-AE and DT-AE (frozen), show positive results compared to no-context BC baselines (in Table 2), BDT outperforms all other learned $\Phi$ variants or Meta-BC and is comparable to CDT or DT (also in Table 2) with longer input ($N = 50$). This implies that even though we don't assume the state-feature specification, aggregator choice in GDT with minimal architectural changes may solve the offline distribution matching problem efficiently. We leave more sophisticated objectives or architectural changes as future work.

| Method | halfcheetah | | | hopper | | | walker2d | | | Average |
|---|---|---|---|---|---|---|---|---|---|---|
| | Expert | Medium | Total | Expert | Medium | Total | Expert | Medium | Total | |
| DT-AE | $2.060 \pm 1.076$ | $1.089 \pm 0.827$ | $1.574 \pm 1.075$ | $0.611 \pm 0.060$ | $0.142 \pm 0.049$ | $0.377 \pm 0.241$ | $0.833 \pm 0.346$ | $0.321 \pm 0.126$ | $0.577 \pm 0.365$ | 0.843 |
| DT-CPC | $6.011 \pm 0.324$ | $1.403 \pm 1.384$ | $3.707 \pm 2.514$ | $0.610 \pm 0.019$ | $0.101 \pm 0.024$ | $0.355 \pm 0.256$ | $1.281 \pm 0.022$ | $0.138 \pm 0.040$ | $0.710 \pm 0.572$ | 1.591 |
| DT-AE (joint) | $8.643 \pm 0.679$ | $2.260 \pm 0.690$ | $5.451 \pm 3.264$ | $1.255 \pm 0.363$ | $0.649 \pm 0.241$ | $0.952 \pm 0.432$ | $2.104 \pm 0.620$ | $0.988 \pm 0.413$ | $1.546 \pm 0.767$ | 2.650 |
| DT-CPC (joint) | $4.544 \pm 1.184$ | $1.884 \pm 1.465$ | $3.214 \pm 1.882$ | $0.575 \pm 0.032$ | $0.096 \pm 0.028$ | $0.335 \pm 0.241$ | $0.949 \pm 0.484$ | $0.412 \pm 0.378$ | $0.680 \pm 0.510$ | 1.410 |
| DT-E2E | $8.208 \pm 1.087$ | $1.928 \pm 0.838$ | $5.068 \pm 3.287$ | $1.097 \pm 0.217$ | $0.542 \pm 0.187$ | $0.820 \pm 0.344$ | $2.086 \pm 0.437$ | $1.239 \pm 0.712$ | $1.662 \pm 0.727$ | 2.517 |
| DT-AE (frozen) | $1.821 \pm 0.582$ | $1.319 \pm 0.863$ | $1.570 \pm 0.778$ | $0.634 \pm 0.038$ | $0.137 \pm 0.062$ | $0.385 \pm 0.253$ | $1.180 \pm 0.328$ | $0.404 \pm 0.170$ | $0.792 \pm 0.468$ | 0.916 |
| DT-CPC (frozen) | $3.489 \pm 1.159$ | $2.543 \pm 0.903$ | $3.016 \pm 1.141$ | $0.631 \pm 0.091$ | $0.171 \pm 0.130$ | $0.401 \pm 0.256$ | $1.315 \pm 0.329$ | $0.279 \pm 0.127$ | $0.797 \pm 0.575$ | 1.405 |
| BDT ($N$=20) | $1.592 \pm 0.201$ | $1.208 \pm 1.854$ | $1.400 \pm 1.333$ | $0.318 \pm 0.093$ | $0.081 \pm 0.013$ | $0.200 \pm 0.136$ | $0.196 \pm 0.031$ | $0.392 \pm 0.184$ | $0.294 \pm 0.164$ | 0.631 |
| BDT ($N$=50) | $0.840 \pm 0.063$ | $1.223 \pm 1.828$ | $1.031 \pm 1.307$ | $0.142 \pm 0.010$ | $0.098 \pm 0.025$ | $0.120 \pm 0.029$ | $0.192 \pm 0.051$ | $0.163 \pm 0.027$ | $0.178 \pm 0.043$ | 0.443 |

Table 5: The results of BDT on the state-feature distribution matching problem ($m = 16$). While even simple auto-encoder regularizers sometimes work well, BDT with longer contexts seems to outperform other strategies and is comparable to CDT or DT (in Table 2). See Appendix E.6 for the full results including $m = 1, 4$.

## 7 CONCLUSION

We provide a unified perspective on a wide range of *hindsight* algorithms, and generalize the problem formulation as hindsight information matching (HIM). Inspired by recent successes in RL as sequence modeling, we propose Generalized Decision Transformer (GDT) which includes DT, Categorical DT, and Bi-directional DT as special cases, and is applicable to any HIM with proper choices of $\Phi$ and aggregator. We show how Categorical DT, a minor modification of DT, enables the first effective offline state-marginal matching algorithm and propose new benchmark tasks for this problem class. We also demonstrate the effectiveness of Bi-directional DT as a one-shot imitation learner, significantly outperforming simple variants based on DT. We hope our proposed HIM and GDT frameworks shed new perspectives on hindsight algorithms and the applicability of sequence modeling to much broader classes of RL problems beyond classic reward-based RL.

ETHICS STATEMENT

Since this paper mainly focuses on the reinterpretation of hindsight RL algorithms and experiments on existing benchmark datasets, we believe there are no ethical concerns.

REPRODUCIBILITY STATEMENT

We share our codes to ensure the reproductivity. The details of hyperparameters are described in Appendix A and B. The detailed settings of experiments are described in the Section 6, Appendix E, and G. We report the results averaged over 20 rollouts every 4 random seed.

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

APPENDIX

## A  HYPER-PARAMETER OF GENERALIZED DECISION TRANSFORMERS

We implement Categorical Decision Transformer and Bi-directional Decision Transformer, built upon the official codebase released by Chen et al. (2021a) (`https://github.com/kzl/decision-transformer`). We follow the most of hyperparameters as they did (Table 6).

| Hyperparameter | Value |
|---|---|
| Number of layers | 3 |
| Number of attention heads | 1 |
| Embedding dimension | 128 |
| Nonlinearity function | ReLU |
| Batch size | 64 |
| Context length $N$ | 20 |
| Dropout | 0.1 |
| Learning rate | 1e-4 |
| Grad norm clip | 0.25 |
| Weight decay | 1e-4 |
| Learning rate decay | Linear warmup for first 100k training steps |
| Training (Gradient) steps | 1M |
| Number of bins for categorical distribution | 31 |
| Encoder size (for DT-$X$) | 2 layer MLP, (128, 128) |
| Coefficient of unsupervised loss | 0.1 |

Table 6: List of hyperparameters for DT, CDT and BDT. We refer Chen et al. (2021a).

## B  DETAILS OF BASELINES

While, to the best of our knowledge, there are no prior work to tackle the offline state-marginal matching problem, we can regard meta or one-shot imitation learning as the methods solving similar problem (see Appendix D for further discussion). In this work, we choose Meta-BC and FOCAL (Li et al., 2021), a metric-based offline meta RL method, as decent baselines.

**FOCAL**   FOCAL is a offline meta RL method combining BRAC (Wu et al., 2019) with metric-based approach. It utilizes deterministic context encoder trained with inverse-power distance metric losses and detached from Bellman backup gradients. We follow the hyperparameters in the official implementation (`https://github.com/LanqingLi1993/FOCAL-ICLR`). Deterministic context encoder takes single-step state-action-reward tuple as a context for task inference. We parameterize it with (200, 200, 200)-layers MLP. We train deterministic context encoder with 100000 iteration first, and then train BRAC agent with task-conditioned policy and value functions with 100000 iteration. We use (256, 256, 256)-layers MLPs for policy and value network, and set batch size to 256.

**Meta-BC**   We also use metric-based Meta-BC (Duan et al., 2017; Dasari & Gupta, 2020) as a strong baseline method for offline multi-task SMM. We adapt deterministic context encoder from FOCAL to infer the task. We train Meta-BC in the same way as FOCAL just replacing BRAC to BC. The objective is mean-squared error minimization.

Throughout our experiments, Meta-BC shows better results than FOCAL, because RL algorithms often struggle to optimize Eq. 4 for distribution matching and FOCAL tries to maximize the task reward, which is not necessarily required for distribution matching problem. In addition, BC often converges faster than offline RL methods.

## C  DETAILS OF EVALUATION

Throughout the paper, we evaluate both CDT and BDT from a distribution matching perspective by formulating them as offline multi-task SMM or offline multi-task IL problem. However, the current distribution matching research in RL (Lee et al., 2020; Ghasemipour et al., 2020; Gu et al., 2021) has been suffered from the lack of quantitative metrics for evaluation (while standard RL or BC are typically evaluated on the task reward performance). As summarized in Table 7, prior works (Lee et al., 2020; Ghasemipour et al., 2020) evaluate the SMM performance by qualitative density visualization using rollout particles.

Although the distance between state-marginal and target distribution seems the most intuitive and suitable metric to quantify the performance of distribution matching algorithms, it is often intractable to measure such distance analytically because both state-marginal and target distribution can be non-parametric; we cannot access their densities, and only their samples are available. While a concurrent work (Gu et al., 2021) tackles this problem by leveraging sample-based energy distance estimation, we introduce a single SMM-inspired evaluation for both offline multi-task SMM and offline multi-task IL via estimating Wasserstein-1 distance between empirical categorical distributions. Since the discretization of low-dimensional features, such as reward, have success in many RL methods (Bellemare et al., 2017; Dabney et al., 2018; 2020; Furuta et al., 2021b), quantification of distribution matching performance by Wasserstein-1 distance could be reliable evaluations (in addition Wasserstein-1 distance is symmetric, different from KL distance that is asymmetric). We note that due to the equivalence between inverse RL and SMM methods, the performance of distribution matching methods may be measured via task reward when assuming the accessivity to the expert trajectories (Ho & Ermon, 2016; Fu et al., 2018; Kostrikov et al., 2019). However, in our experiments, it is not suitable to evaluate the performance based on task reward since we uses the sub-optimal trajectories (Section 6.1) or multi-task trajectories from different reward functions (Section 6.2).

Wasserstein-1 distance between state-marginal distribution $\rho^\pi(s)$ and target distribution $p^*(s)$ is defined as:

$$W_1(\rho^\pi(s), p^*(s)) = \inf_{\nu \in \Gamma(\rho^\pi, p^*)} \mathbb{E}_{(X,Y) \sim \nu}[||X - Y||], \tag{5}$$

where $\Gamma$ is the set of all possible joint distributions $\nu$ whose marginals are $\rho^\pi$ and $p^*$ respectively ($X$ and $Y$ are random variables). In this work, we empirically estimate Eq. 5 by focusing on the "manually-specified" feature $\phi \in F$ (e.g. reward, xyz-velocities, etc; defined in Section 4) and employing binning and discretization. We discretize the feature space $F$ and obtain $B$ bins (per dimension), whose representatives are $\{\bar{\phi}_1, \bar{\phi}_2, \ldots \bar{\phi}_B\}$. The range of feature space $[\phi_{\min}, \phi_{\max}]$ is pre-defined from the given offline datasets. Then, we get categorical distribution from the target trajectory $\tau$; $\hat{p}_\tau^*(\phi) = \{(\bar{\phi}_1, c_{\tau 1}^*), \ldots (\bar{\phi}_B, c_{\tau B}^*)\}$, and from the rollouts of the policy $\pi$ (using 4 seeds $\times$ 20 rollouts); $\hat{\rho}^\pi(\phi) = \{(\bar{\phi}_1, c_1^\pi), \ldots (\bar{\phi}_B, c_B^\pi)\}$, where $c_\tau^*$ and $c^\pi$ are the weights of each bin (i.e. $\sum_{l=1}^B c_{\tau l}^* = 1$ and $\sum_{l=1}^B c_l^\pi = 1$). We compute the evaluation metric averaging on the test set of the trajectories $D_{\text{test}}$:

$$\text{Metric}(\pi, D_{\text{test}}) = \frac{1}{|D_{\text{test}}|} \sum_{\tau \in D_{\text{test}}} \min_{w_{ij}} \sum_{i=1}^B \sum_{j=1}^B w_{ij} |c_{\tau i}^* - c_j^\pi|,$$

$$\text{s.t. } w_{ij} \geq 0, \ \sum_{j=1}^B w_{ij} \leq c_{\tau i}^*, \ \sum_{i=1}^B w_{ij} \leq c_j^\pi, \ \sum_{i=1}^B \sum_{j=1}^B w_{ij} = 1. \tag{6}$$

In practice, we utilize the python package (https://github.com/pkomiske/Wasserstein) released by Komiske et al. (2020) to compute it.

| Evaluation | Type | Reference |
|---|---|---|
| Density Visualization | Qualitative | Lee et al. (2020); Ghasemipour et al. (2020) |
| Energy Distance | Quantitative | Gu et al. (2021) |
| Task Reward | Quantitative | Ho & Ermon (2016); Fu et al. (2018), etc. |
| Wasserstein-1 Distance | Quantitative | Ours |

Table 7: A Review of evaluation for distribution matching algorithms.

# D    CONNECTION TO META LEARNING

While we solve offline information matching problems in this paper, our formulations (especially Bi-directional DT) and experimental settings are similar to offline meta RL (Dorfman et al., 2021; Mitchell et al., 2021; Li et al., 2021) and meta/one-shot imitation learning (Yu et al., 2019; Ghasemipour et al., 2019; Finn et al., 2017; Duan et al., 2017) (especially metric-based approach (Rakelly et al., 2019; Duan et al., 2016; Fakoor et al., 2020)). We briefly clarify the relevance and difference between them (Table 8).

**Offline Meta RL**  Recently, some works deal with offline meta RL problem, assuming task distribution and offline training with pre-stored transitions collected by the (sub-) optimal or scripted task-conditioned agents (Dorfman et al., 2021; Mitchell et al., 2021; Li et al., 2021; Pong et al., 2021; Zhao et al., 2021). In these works, few test-task trajectories, following the same task distribution but unseen during training phase, are given at test-time (i.e. few-shot), and the agents adapt the given task in an online (Pong et al., 2021; Zhao et al., 2021; Dorfman et al., 2021) or fully-offline (Mitchell et al., 2021; Li et al., 2021) manner.

**Meta Imitation Learning**  Another related domain is meta imitation learning, also assuming task distribution and offline training with pre-stored expert transitions per tasks (Yu et al., 2019; Ghasemipour et al., 2019; Finn et al., 2017; Duan et al., 2017; Xu et al., 2019; Gleave & Habryka, 2018; Dasari & Gupta, 2020). While meta inverse RL methods (Yu et al., 2019; Xu et al., 2019; Gleave & Habryka, 2018) require online samples during test-time adaptation, its off-policy extension and BC-based approach performs offline adaptation with given unseen trajectories (Ghasemipour et al., 2019; Finn et al., 2017; Dasari & Gupta, 2020).

Our work, in contrast, assume no-adaptation at test time and evaluation criteria is a distance between the feature distributions, not the task rewards.

| Method | Problem | Train | Test | Demo |
|---|---|---|---|---|
| Pong et al. (2021) | Offline Meta RL | offline | online | few-shot |
| Zhao et al. (2021) | Offline Meta RL | offline | online | few-shot |
| Dorfman et al. (2021) | Offline Meta RL | offline | online | few-shot |
| Mitchell et al. (2021) | Offline Meta RL | offline | offline | few-shot |
| Li et al. (2021) | Offline Meta RL | offline | offline | few-shot |
| Yu et al. (2019) | Meta IL | online | online | few-shot |
| Xu et al. (2019) | Meta IL | online | online | few-shot |
| Ghasemipour et al. (2019) | Meta IL | online | offline | few-shot |
| Finn et al. (2017) | Meta IL | offline | offline | one-shot |
| Duan et al. (2017) | Meta IL | offline | (no-adaptation) | one-shot |
| Ours | Offline multi-task SMM/IL | offline | (no-adaptation) | one-shot |

Table 8: Review of problem settings among offline meta RL, meta IL, and ours. In offline meta RL, the agents are trained offline and tested with several demonstrations (few-shot), in a fully-offline manner or allowing online data-collection for adaptation. In meta IL, IRL-based methods train the agents online (Yu et al., 2019; Xu et al., 2019; Ghasemipour et al., 2019) while BC-based methods (Finn et al., 2017; Duan et al., 2017) do offline. Similar to our settings, Duan et al. (2017) test the agents with single demonstration and no-adaptation process, they evaluate the agent on the task reward, while we do on the distance between the two distributions.

# E   DETAILS OF EXPERIMENTS

In this section, we provide the details and supplemental quantitative/qualitative results of the experiments in Section 6.

Figure 3 visualizes the dataset distributions we use in the experiments. When we sort all the trajectories based on their cumulative rewards, each quality of trajectory (best, middle, worst) shows a different shape of distribution, and the distribution of the whole dataset seems a weighted mixture of those.

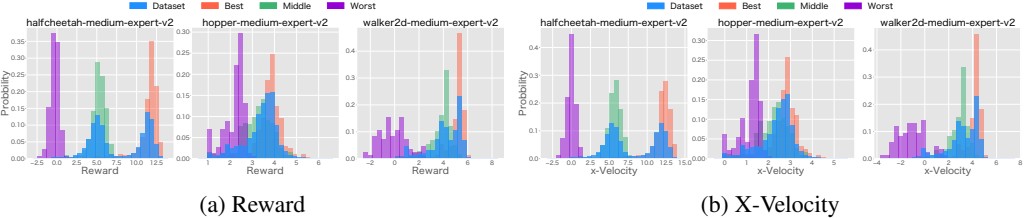

(a) Reward                                      (b) X-Velocity

Figure 3: Distributions of the features (reward, and x-velocity) in the D4RL medium-expert datasets.

## E.1   QUANTITATIVE AND QUALITATIVE RESULTS OF REWARD AND STATE-FEATURE MATCHING

We provide the quantitative comaprison of Section 6.1 between Categorical DT and DT in the reward (Table 9) matching problem, computing Wasserstein-1 distance between the discretized rollout and target feature distributions. Generally, similar to the x-velocity case, CDT shows the better results in offline multi-task SMM compared to original DT. We also visualize some of CDT results in Figure 4.

| Method | halfcheetah | | | hopper | | | walker2d | | | Average |
| --- | --- | --- | --- | --- | --- | --- | --- | --- | --- | --- |
| | Expert | Medium | Total | Expert | Medium | Total | Expert | Medium | Total | |
| Categorical DT | 0.674 ± 0.213 | 1.002 ± 1.458 | 0.838 ± 1.054 | 0.159 ± 0.085 | 0.064 ± 0.017 | 0.111 ± 0.077 | 0.095 ± 0.017 | 0.114 ± 0.037 | 0.105 ± 0.030 | 0.351 |
| DT | 0.652 ± 0.319 | 1.039 ± 1.548 | 0.846 ± 1.134 | 0.227 ± 0.119 | 0.091 ± 0.035 | 0.159 ± 0.111 | 0.056 ± 0.015 | 0.626 ± 0.495 | 0.341 ± 0.452 | 0.448 |
| BC (no-context) | 3.240 ± 0.559 | 2.880 ± 0.614 | 3.060 ± 0.614 | 0.597 ± 0.056 | 0.119 ± 0.067 | 0.358 ± 0.247 | 0.977 ± 0.501 | 0.431 ± 0.396 | 0.704 ± 0.528 | 1.374 |
| Meta-BC | 0.839 ± 0.682 | 0.830 ± 1.130 | 0.835 ± 0.933 | 0.803 ± 0.505 | 0.134 ± 0.056 | 0.468 ± 0.491 | 0.113 ± 0.085 | 1.441 ± 1.113 | 0.777 ± 1.032 | 0.693 |
| FOCAL (Li et al., 2021) | 1.623 ± 0.501 | 1.115 ± 1.534 | 1.369 ± 0.516 | 1.463 ± 0.472 | 0.492 ± 0.384 | 0.977 ± 0.649 | 1.584 ± 0.570 | 0.604 ± 0.421 | 1.094 ± 0.701 | 1.147 |

Table 9: Quantitative evaluation of reward distribution matching via measuring Wasserstein-1 distance between the rollout and target distributions. We compare Categorical DT and DT. Since it can capture the multi-modal nature of target distributions, CDT matches the distribution better than the original DT.

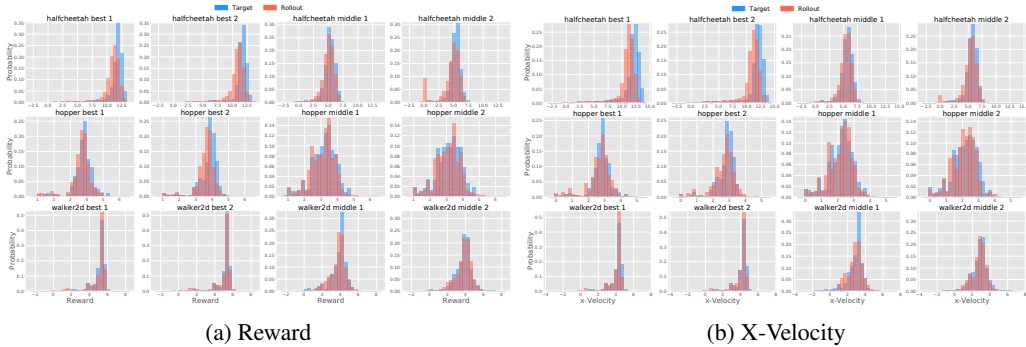

(a) Reward                                      (b) X-Velocity

Figure 4: (a) Reward and (b) state-feature (x-velocity) distribution matching in halfcheetah (top), hopper (middle), and walker2d (bottom). The left two examples are the distributions from the best trajectories, and right two are the distributions from the middle trajectories in the held-out test set. The rollout distributions of CDT (red) match the target distributions (blue) very well in all cases.

## E.2 EVALUATION ON TASK PERFORMANCE

While in this paper we focus on the evaluation with the SMM-inspired distribution matching objective, such as Wasserstein-1 distance, we here provide the evaluation on the task rewards. Table 10 shows that DT seems consistently better methods than Categorical DT on the task rewards evaluations, and achieves similar performances to the held-out trajectories.

| Method | halfcheetah | | hopper | | walker2d | |
|---|---|---|---|---|---|---|
| | Expert | Medium | Expert | Medium | Expert | Medium |
| Categorical DT | $10476.746 \pm 218.957$ | $5782.557 \pm 1666.716$ | $2614.559 \pm 657.722$ | $1518.757 \pm 63.062$ | $4907.475 \pm 11.711$ | $3264.585 \pm 209.905$ |
| DT | $10500.273 \pm 312.778$ | $4985.518 \pm 67.322$ | $2113.207 \pm 807.246$ | $1528.743 \pm 30.780$ | $4965.024 \pm 11.814$ | $4055.039 \pm 849.109$ |
| Held-out | $11146.200 \pm 59.070$ | $5237.348 \pm 37.970$ | $3741.854 \pm 7.223$ | $1600.196 \pm 0.772$ | $4995.553 \pm 5.413$ | $3801.313 \pm 0.994$ |

Table 10: Evaluation on the task rewards; conditioning on the held-out trajectories as done in Section 6. We compare the performance between Categorical DT, and DT. DT seems consistently better methods on the task rewards evaluations, and achieves similar performances to the held-out trajectories (averaged over 5 trajectories).

## E.3 2D STATE-FEATURE DISTRIBUTION MATCHING

We also consider two-dimensional state-features (xy-velocities) distribution matching in Ant-v3. Same as 1D state-feature distribution matching in HalfCheetah, Hopper, and Walker2d-v3 (Section 6.1), we also use medium-expert(-v2) datasets from D4RL (Fu et al., 2020). We bin the state-features per dimension separately to reduce the dimension of the categorical distribution that CDT takes as input, while in test-time we evaluate the performance with Wasserstein-1 metric on the joint distribution. For DT, we compute the summation of x- and y-velocity each over trajectories and normalize them with the maximum horizon. DT feeds these two scalars as information statistics to match.

Table 11 reveals that CDT performs better even in the case of two-dimensional state-features distributions, while DT doesn't generalize to expert-quality trajectories. As shown in Figure 5, while CDT could cope with the distribution shift between expert and medium target distribution, DT always fits the medium one even if the expert trajectory is given as a target. CDT successfully scales to the offline multi-task SMM problem in the multi-dimensional feature spaces.

| Method | ant | | |
|---|---|---|---|
| | Expert | Medium | Average |
| Categorical DT | $0.797 \pm 0.216$ | $0.244 \pm 0.063$ | 0.521 |
| DT | $1.714 \pm 0.121$ | $0.260 \pm 0.067$ | 0.987 |
| Meta-BC | $1.295 \pm 0.708$ | $0.351 \pm 0.205$ | 0.823 |
| FOCAL (Li et al., 2021) | $1.473 \pm 0.892$ | $0.913 \pm 0.455$ | 1.193 |

Table 11: Quantitative evaluation of 2D state-feature (xy-velocities) distribution matching, measuring Wasserstein-1 distance. CDT performs better even in the two-dimensional problem.

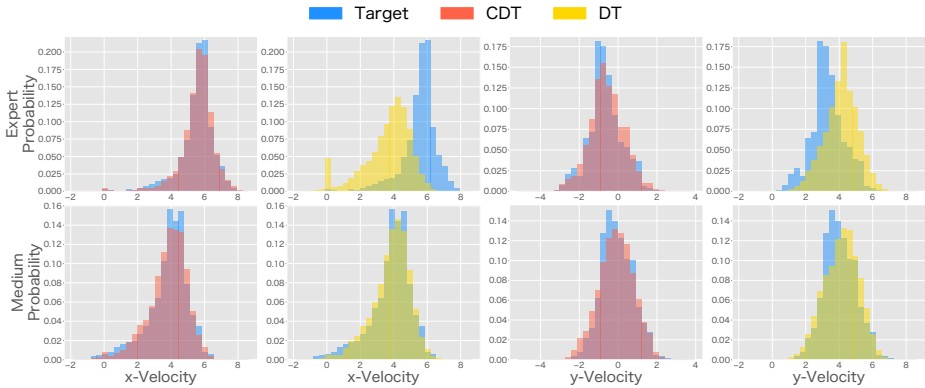

Figure 5: Visualization of 2D state-feature (xy-velocities) distribution matching, binning each dimension separately. Top row shows the results from an expert target trajectory, and bottom row from a medium one.

### E.4 DETAILS OF SYNTHESIZING UNSEEN BI-MODAL DISTRIBUTION

To construct the dataset, we modified the original reward function in HalfCheetah-v3, adding absolute z-velocity term (such as `+np.abs(z_vel)`), where the expert cheetah backflips towards -x direction.

We trained SAC agent until convergence (3 million gradient steps), using pytorch implementation released by Furuta et al. (2021a), and then collected 500 trajectories $\times$ 1000 time steps. We combined them with 500 trajectories $\times$ 1000 time steps from halfcheetah-expert-v2 dataset in D4RL, which consists of both backflipping and running forward behaviors.

For the evaluation, we prepared additional 5 trajectories of backflipping and 5 of running forward as uni-modal behaviors (10 test trajectories in total). In addition, we synthesized a bi-modal behavior by dividing each 1000-step trajectories into 500-step sub-trajectories, and concatenating them across different behaviors, which results in the patchworked trajectories of first 500-step running forward and next 500-step backflipping (also 10 trajectories in total).

### E.5 DETAILS OF DIVERSE UNSEEN DISTRIBUTION FROM META LEARNING TASK

Following prior meta RL/IL works (Rakelly et al., 2019; Ghasemipour et al., 2019; Pong et al., 2021; Li et al., 2021; Fakoor et al., 2020), we modified the reward function for the cheetah to run with specified velocity (such as `-np.abs(x_vel - target_vel)`), and set the horizon to 200 steps.

We prepared 31 target x-velocities; taken from [0.0, 3.0], uniformly spaced at 0.1 intervals. We also trained the SAC agents until convergence (3 million gradient steps), using pytorch implementation released by Furuta et al. (2021a), and collected 250 trajectories $\times$ 200 time steps each. To simplify the problem than meta learning settings, we held out the 10 trajectories whose x-velocity is $\{0.5, 1.5, 2.5\}$ as a test set, and used the rest as a train data.

### E.6 QUANTITATIVE AND QUALITATIVE RESULTS OF ONE-SHOT DISTRIBUTION MATCHING IN FULL STATE

Table 12 and Table 13 show the one-shot reward and state-feature distribution matching results respectively. In addition to the embedding size $m$, we also sweep the different context window $N = 20, 50, 100$ for BDT ($m = 16$). While even simple auto-encoder regularizer (DT-AE or DT-AE (frozen)) sometimes works well compared to no-context BC baselines presented in Table 2 and Table 9, BDT, using (second) anti-causal transformer as an encoder $\Phi$ and its aggregator, seems consistently better than other strategies or Meta-BC baseline for offline multi-task SMM and is comparable to CDT or DT results (also presented in Table 2 and Table 9) with longer context length ($N = 50$). Through the experiment we observe that while there are no clear trends in DT-AE, -CPC or -E2E as the size of context embedding $m$ grows, BDT improves its performance with larger size of embedding. Such intuitive properties might a good features to design the architectures.

We visualize the qualitative results of BDT ($m = 16, N = 20$) in Figure 6.

| Method | halfcheetah | | | hopper | | | walker2d | | | Average |
|---|---|---|---|---|---|---|---|---|---|---|
| | Expert | Medium | Total | Expert | Medium | Total | Expert | Medium | Total | |
| DT-AE (m=1) | 2.494 ± 1.050 | 2.877 ± 0.762 | 2.686 ± 0.937 | 0.586 ± 0.016 | 0.089 ± 0.029 | 0.337 ± 0.249 | 0.733 ± 0.522 | 0.586 ± 0.445 | 0.660 ± 0.490 | 1.228 |
| DT-CPC (m=1) | 3.595 ± 0.676 | 2.275 ± 0.341 | 2.935 ± 0.849 | 0.600 ± 0.007 | 0.130 ± 0.039 | 0.365 ± 0.237 | 1.180 ± 0.019 | **0.139 ± 0.032** | 0.660 ± 0.521 | 1.320 |
| DT-AE (m=4) | 0.782 ± 0.329 | 1.720 ± 1.685 | 1.251 ± 1.301 | 0.613 ± 0.108 | 0.140 ± 0.068 | 0.376 ± 0.253 | 0.889 ± 0.402 | 0.265 ± 0.137 | 0.577 ± 0.433 | 0.735 |
| DT-CPC (m=4) | 5.887 ± 0.357 | 1.370 ± 1.338 | 3.628 ± 2.462 | 0.737 ± 0.018 | 0.238 ± 0.060 | 0.487 ± 0.254 | 1.286 ± 0.018 | 0.145 ± 0.031 | 0.715 ± 0.571 | 1.610 |
| DT-AE (m=16) | 2.041 ± 1.080 | 1.074 ± 0.814 | 1.558 ± 1.071 | 0.613 ± 0.060 | 0.146 ± 0.051 | 0.379 ± 0.240 | 0.837 ± 0.345 | 0.324 ± 0.126 | 0.581 ± 0.365 | 0.839 |
| DT-CPC (m=16) | 6.022 ± 0.316 | 1.406 ± 1.371 | 3.714 ± 2.513 | 0.614 ± 0.019 | 0.104 ± 0.025 | 0.359 ± 0.256 | 1.284 ± 0.020 | 0.140 ± 0.039 | 0.712 ± 0.572 | 1.595 |
| DT-AE (m=1, joint) | 3.824 ± 1.114 | 1.988 ± 0.970 | 2.906 ± 1.391 | 0.687 ± 0.097 | 0.137 ± 0.057 | 0.412 ± 0.286 | 0.723 ± 0.596 | 0.614 ± 0.462 | 0.668 ± 0.536 | 1.329 |
| DT-CPC (m=1, joint) | 4.460 ± 1.106 | 2.036 ± 1.032 | 3.248 ± 1.616 | 0.587 ± 0.015 | 0.106 ± 0.042 | 0.347 ± 0.243 | 0.815 ± 0.711 | 0.710 ± 0.398 | 0.763 ± 0.578 | 1.452 |
| DT-AE (m=4, joint) | 5.563 ± 0.449 | 1.028 ± 1.265 | 3.295 ± 2.458 | 1.320 ± 0.208 | 0.485 ± 0.235 | 0.902 ± 0.473 | 0.917 ± 0.319 | 0.218 ± 0.112 | 0.567 ± 0.423 | 1.588 |
| DT-CPC (m=4, joint) | 3.486 ± 1.503 | 2.265 ± 1.077 | 2.876 ± 1.443 | 0.690 ± 0.159 | 0.220 ± 0.176 | 0.455 ± 0.288 | 1.021 ± 0.708 | 0.554 ± 0.383 | 0.788 ± 0.615 | 1.373 |
| DT-AE (m=16, joint) | 8.450 ± 0.623 | 2.168 ± 0.701 | 5.309 ± 3.210 | 1.257 ± 0.361 | 0.655 ± 0.242 | 0.956 ± 0.430 | 2.112 ± 0.618 | 0.994 ± 0.413 | 1.553 ± 0.767 | 2.606 |
| DT-CPC (m=16, joint) | 4.543 ± 1.179 | 1.869 ± 1.453 | 3.206 ± 1.881 | 0.577 ± 0.032 | 0.098 ± 0.028 | 0.338 ± 0.241 | 0.953 ± 0.483 | 0.414 ± 0.376 | 0.683 ± 0.510 | 1.409 |
| DT-E2E (m=1) | 3.220 ± 2.325 | 1.026 ± 1.398 | 2.123 ± 2.210 | 1.615 ± 0.536 | 0.540 ± 0.293 | 1.078 ± 0.690 | 1.079 ± 0.209 | 0.455 ± 0.085 | 0.767 ± 0.351 | 1.323 |
| DT-E2E (m=4) | 8.076 ± 0.551 | 2.552 ± 0.571 | 5.314 ± 2.818 | 1.205 ± 0.390 | 0.493 ± 0.247 | 0.849 ± 0.483 | 2.835 ± 0.946 | 1.239 ± 0.460 | 2.037 ± 1.091 | 2.733 |
| DT-E2E (m=16) | 8.049 ± 0.990 | 1.859 ± 0.839 | 4.954 ± 3.228 | 1.102 ± 0.216 | 0.549 ± 0.189 | 0.826 ± 0.343 | 2.095 ± 0.434 | 1.241 ± 0.709 | 1.668 ± 0.726 | 2.482 |
| DT-AE (m=1, frozen) | 2.225 ± 1.017 | 2.804 ± 1.051 | 2.514 ± 1.074 | 0.582 ± 0.042 | 0.131 ± 0.058 | 0.357 ± 0.231 | 1.396 ± 0.149 | 0.259 ± 0.083 | 0.827 ± 0.581 | 1.233 |
| DT-CPC (m=1, frozen) | 4.110 ± 0.799 | 2.172 ± 0.789 | 3.141 ± 1.253 | 0.582 ± 0.021 | 0.102 ± 0.041 | 0.342 ± 0.242 | 1.422 ± 0.099 | 0.275 ± 0.109 | 0.848 ± 0.583 | 1.444 |
| DT-AE (m=4, frozen) | 0.815 ± 0.183 | 1.415 ± 1.755 | 1.115 ± 1.283 | 0.681 ± 0.106 | 0.197 ± 0.061 | 0.439 ± 0.257 | 1.066 ± 0.495 | 0.419 ± 0.321 | 0.742 ± 0.528 | 0.765 |
| DT-CPC (m=4, frozen) | 3.275 ± 1.186 | 2.752 ± 1.088 | 3.014 ± 1.168 | 0.633 ± 0.055 | 0.139 ± 0.082 | 0.386 ± 0.257 | 1.119 ± 0.598 | 0.508 ± 0.354 | 0.814 ± 0.578 | 1.404 |
| DT-AE (m=16, frozen) | 1.796 ± 0.578 | 1.312 ± 0.852 | 1.554 ± 0.767 | 0.637 ± 0.039 | 0.142 ± 0.063 | 0.389 ± 0.253 | 1.184 ± 0.326 | 0.405 ± 0.169 | 0.795 ± 0.469 | 0.913 |
| DT-CPC (m=16, frozen) | 3.486 ± 1.164 | 2.538 ± 0.905 | 3.012 ± 1.145 | 0.636 ± 0.093 | 0.178 ± 0.132 | 0.407 ± 0.256 | 1.318 ± 0.328 | 0.282 ± 0.126 | 0.800 ± 0.574 | 1.407 |
| BDT (m=1, N=20) | 1.385 ± 0.207 | 1.180 ± 1.753 | 1.282 ± 1.253 | 0.291 ± 0.096 | 0.110 ± 0.043 | 0.201 ± 0.117 | 1.113 ± 0.044 | 0.155 ± 0.046 | 0.634 ± 0.481 | 0.706 |
| BDT (m=4, N=20) | 1.660 ± 0.167 | 1.058 ± 1.580 | 1.359 ± 1.163 | 0.494 ± 0.281 | 0.096 ± 0.029 | 0.295 ± 0.282 | 0.181 ± 0.023 | 0.414 ± 0.537 | 0.298 ± 0.397 | 0.650 |
| BDT (m=16, N=20) | 1.565 ± 0.190 | 1.191 ± 1.830 | 1.378 ± 1.315 | 0.321 ± 0.093 | 0.086 ± 0.017 | 0.204 ± 0.135 | 0.204 ± 0.033 | 0.396 ± 0.186 | 0.300 ± 0.165 | 0.627 |
| BDT (m=16, N=50) | 0.831 ± 0.064 | 1.204 ± 1.803 | 1.018 ± 1.290 | 0.144 ± 0.011 | 0.104 ± 0.026 | 0.124 ± 0.028 | 0.199 ± 0.052 | 0.167 ± 0.024 | 0.183 ± 0.044 | 0.442 |
| BDT (m=16, N=100) | 0.936 ± 0.166 | 1.280 ± 1.861 | 1.108 ± 1.332 | 0.240 ± 0.047 | 0.162 ± 0.033 | 0.201 ± 0.056 | 0.057 ± 0.007 | 0.873 ± 0.607 | 0.465 ± 0.593 | 0.591 |

Table 12: The results of BDT and DT-$X$ variants on the reward distribution matching problem ($m = 1, 4, 16$).

| Method | halfcheetah | | | hopper | | | walker2d | | | Average |
|---|---|---|---|---|---|---|---|---|---|---|
| | Expert | Medium | Total | Expert | Medium | Total | Expert | Medium | Total | |
| DT-AE (m=1) | 2.504 ± 1.050 | 2.880 ± 0.778 | 2.692 ± 0.943 | 0.580 ± 0.015 | 0.084 ± 0.027 | 0.332 ± 0.249 | 0.729 ± 0.522 | 0.584 ± 0.447 | 0.656 ± 0.491 | 1.227 |
| DT-CPC (m=1) | 3.595 ± 0.670 | 2.277 ± 0.349 | 2.936 ± 0.848 | 0.601 ± 0.008 | 0.130 ± 0.037 | 0.365 ± 0.237 | 1.177 ± 0.021 | 0.135 ± 0.033 | 0.656 ± 0.522 | 1.319 |
| DT-AE (m=4) | 0.789 ± 0.333 | 1.729 ± 1.714 | 1.259 ± 1.321 | 0.612 ± 0.108 | 0.138 ± 0.066 | 0.375 ± 0.254 | 0.884 ± 0.402 | 0.262 ± 0.138 | 0.573 ± 0.433 | 0.736 |
| DT-CPC (m=4) | 5.883 ± 0.361 | 1.371 ± 1.347 | 3.627 ± 2.462 | 0.731 ± 0.018 | 0.229 ± 0.057 | 0.480 ± 0.255 | 1.282 ± 0.022 | 0.141 ± 0.032 | 0.712 ± 0.571 | 1.606 |
| DT-AE (m=16) | 2.060 ± 1.076 | 1.089 ± 0.827 | 1.574 ± 1.075 | 0.611 ± 0.060 | 0.142 ± 0.049 | 0.377 ± 0.241 | 0.833 ± 0.346 | 0.321 ± 0.126 | 0.577 ± 0.365 | 0.843 |
| DT-CPC (m=16) | 6.011 ± 0.324 | 1.403 ± 1.384 | 3.707 ± 2.514 | 0.610 ± 0.019 | 0.101 ± 0.024 | 0.355 ± 0.256 | 1.281 ± 0.022 | 0.138 ± 0.040 | 0.710 ± 0.572 | 1.591 |
| DT-AE (m=1, joint) | 3.825 ± 1.115 | 1.990 ± 0.988 | 2.908 ± 1.397 | 0.683 ± 0.094 | 0.132 ± 0.055 | 0.407 ± 0.286 | 0.719 ± 0.597 | 0.611 ± 0.463 | 0.665 ± 0.537 | 1.327 |
| DT-CPC (m=1, joint) | 4.460 ± 1.101 | 2.035 ± 1.055 | 3.248 ± 1.622 | 0.583 ± 0.015 | 0.099 ± 0.041 | 0.341 ± 0.244 | 0.811 ± 0.711 | 0.709 ± 0.399 | 0.760 ± 0.579 | 1.450 |
| DT-AE (m=4, joint) | 5.589 ± 0.458 | 1.040 ± 1.270 | 3.315 ± 2.467 | 1.318 ± 0.208 | 0.481 ± 0.234 | 0.899 ± 0.473 | 0.912 ± 0.319 | 0.216 ± 0.112 | 0.564 ± 0.422 | 1.593 |
| DT-CPC (m=4, joint) | 3.484 ± 1.498 | 2.270 ± 1.099 | 2.877 ± 1.448 | 0.685 ± 0.157 | 0.214 ± 0.174 | 0.449 ± 0.288 | 1.063 ± 0.713 | 0.555 ± 0.384 | 0.786 ± 0.618 | 1.371 |
| DT-AE (m=16, joint) | 8.643 ± 0.679 | 2.260 ± 0.690 | 5.451 ± 3.264 | 1.255 ± 0.363 | 0.649 ± 0.241 | 0.952 ± 0.432 | 2.104 ± 0.620 | 0.988 ± 0.413 | 1.546 ± 0.767 | 2.650 |
| DT-CPC (m=16, joint) | 4.544 ± 1.184 | 1.884 ± 1.465 | 3.214 ± 1.882 | 0.575 ± 0.032 | 0.096 ± 0.028 | 0.335 ± 0.241 | 0.949 ± 0.484 | 0.412 ± 0.378 | 0.680 ± 0.510 | 1.410 |
| DT-E2E (m=1) | 3.265 ± 2.346 | 1.050 ± 1.413 | 2.158 ± 2.231 | 1.613 ± 0.540 | 0.534 ± 0.293 | 1.073 ± 0.692 | 1.073 ± 0.211 | 0.451 ± 0.084 | 0.762 ± 0.350 | 1.331 |
| DT-E2E (m=4) | 8.215 ± 0.604 | 2.684 ± 0.575 | 5.449 ± 2.828 | 1.202 ± 0.392 | 0.487 ± 0.246 | 0.845 ± 0.485 | 2.832 ± 0.951 | 1.235 ± 0.463 | 2.034 ± 1.094 | 2.776 |
| DT-E2E (m=16) | 8.208 ± 1.087 | 1.928 ± 0.838 | 5.068 ± 3.287 | 1.097 ± 0.217 | 0.542 ± 0.187 | 0.820 ± 0.344 | 2.086 ± 0.437 | 1.239 ± 0.712 | 1.662 ± 0.727 | 2.517 |
| DT-AE (m=1, frozen) | 2.238 ± 1.011 | 2.807 ± 1.056 | 2.523 ± 1.072 | 0.577 ± 0.043 | 0.126 ± 0.056 | 0.352 ± 0.231 | 1.391 ± 0.149 | 0.256 ± 0.083 | 0.824 ± 0.580 | 1.233 |
| DT-CPC (m=1, frozen) | 4.110 ± 0.794 | 2.173 ± 0.806 | 3.141 ± 1.257 | 0.578 ± 0.022 | 0.097 ± 0.039 | 0.338 ± 0.242 | 1.417 ± 0.100 | 0.272 ± 0.109 | 0.845 ± 0.582 | 1.441 |
| DT-AE (m=4, frozen) | 0.825 ± 0.189 | 1.431 ± 1.782 | 1.128 ± 1.303 | 0.679 ± 0.106 | 0.193 ± 0.060 | 0.436 ± 0.258 | 1.063 ± 0.495 | 0.418 ± 0.322 | 0.740 ± 0.527 | 0.768 |
| DT-CPC (m=4, frozen) | 3.274 ± 1.187 | 2.756 ± 1.094 | 3.015 ± 1.170 | 0.629 ± 0.054 | 0.135 ± 0.081 | 0.382 ± 0.257 | 1.115 ± 0.600 | 0.507 ± 0.353 | 0.811 ± 0.578 | 1.403 |
| DT-AE (m=16, frozen) | 1.821 ± 0.582 | 1.319 ± 0.863 | 1.570 ± 0.778 | 0.634 ± 0.038 | 0.137 ± 0.062 | 0.385 ± 0.253 | 1.180 ± 0.328 | 0.404 ± 0.170 | 0.792 ± 0.468 | 0.916 |
| DT-CPC (m=16, frozen) | 3.489 ± 1.159 | 2.543 ± 0.903 | 3.016 ± 1.141 | 0.631 ± 0.091 | 0.171 ± 0.130 | 0.401 ± 0.256 | 1.315 ± 0.329 | 0.279 ± 0.127 | 0.797 ± 0.575 | 1.405 |
| BDT (m=1, N=20) | 1.414 ± 0.210 | 1.197 ± 1.770 | 1.305 ± 1.265 | 0.288 ± 0.096 | 0.108 ± 0.041 | 0.198 ± 0.116 | 1.108 ± 0.045 | 0.152 ± 0.051 | 0.630 ± 0.480 | 0.711 |
| BDT (m=4, N=20) | 1.694 ± 0.171 | 1.071 ± 1.594 | 1.382 ± 1.175 | 0.490 ± 0.280 | 0.092 ± 0.030 | 0.291 ± 0.281 | 0.173 ± 0.024 | 0.411 ± 0.547 | 0.292 ± 0.405 | 0.655 |
| BDT (m=16, N=20) | 1.592 ± 0.201 | 1.208 ± 1.854 | 1.400 ± 1.333 | 0.318 ± 0.093 | 0.081 ± 0.013 | 0.200 ± 0.136 | 0.196 ± 0.031 | 0.392 ± 0.184 | 0.294 ± 0.164 | 0.631 |
| BDT (m=16, N=50) | 0.840 ± 0.063 | 1.223 ± 1.828 | 1.031 ± 1.307 | 0.142 ± 0.010 | 0.098 ± 0.025 | 0.120 ± 0.029 | 0.192 ± 0.051 | 0.163 ± 0.027 | 0.178 ± 0.043 | 0.443 |
| BDT (m=16, N=100) | 0.953 ± 0.168 | 1.308 ± 1.881 | 1.130 ± 1.347 | 0.240 ± 0.044 | 0.156 ± 0.032 | 0.198 ± 0.057 | 0.051 ± 0.006 | 0.883 ± 0.614 | 0.467 ± 0.601 | 0.598 |

Table 13: The results of BDT and DT-$X$ variants on the state-feature (x-velocity) distribution matching problem ($m = 1, 4, 16$).

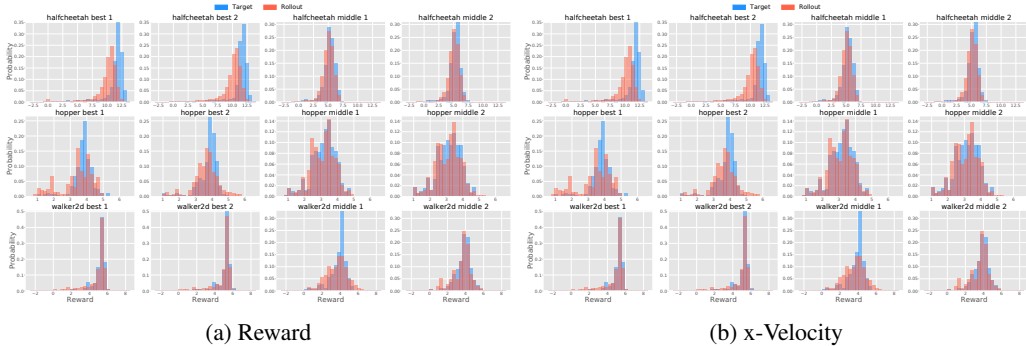

(a) Reward  (b) x-Velocity

Figure 6: (a) Reward and (b) state-feature distribution matching by Bi-directional Decision Transformer ($m = 16$) in halfcheetah (top), hopper (middle), and walker2d (bottom). The left two examples are the distributions from the best trajectories, and right two are the distributions from the middle trajectories.

### E.7 SHIFTING THE TARGET DISTRIBUTION

To generate the unseen but manageable generalization-test trajectories within the support of dataset distribution, we make the reward and velocities values of trajectories in the test set shift with a constant offset: `bin_size` $\times\{-3.0, -2.0, -1.0, 0.0, +1.0, +2.0, +3.0\}$. Table 14 shows the quantitative comparison between CDT and DT, based on Wasserstein-1 distance between two distributions. CDT successfully handles the distribution shifts (especially in hopper) better than DT. We also provide the state-feature results (Table 15) and qualitative visualizations (Figure 7 and Figure 8), which reveals that CDT can match the rollouts to the shifted target distributions when they are within the support of dataset distributions.

| Method | halfcheetah | | | hopper | | | walker2d | | | Average |
|---|---|---|---|---|---|---|---|---|---|---|
| | Expert | Medium | Total | Expert | Medium | Total | Expert | Medium | Total | |
| Categorical DT | $1.126 \pm 0.245$ | $2.026 \pm 1.180$ | $1.576 \pm 0.964$ | $0.147 \pm 0.034$ | $0.302 \pm 0.085$ | $0.224 \pm 0.101$ | $0.285 \pm 0.044$ | $1.024 \pm 0.076$ | $0.655 \pm 0.375$ | 0.818 |
| DT | $1.133 \pm 0.197$ | $1.978 \pm 1.104$ | $1.555 \pm 0.899$ | $0.521 \pm 0.041$ | $0.531 \pm 0.045$ | $0.526 \pm 0.043$ | $0.656 \pm 0.380$ | $0.915 \pm 0.106$ | $0.786 \pm 0.308$ | 0.956 |

Table 14: Wasserstein-1 distance between shifted reward distribution and the rollout distributions. Categorical DT handles the target distribution shifts and matches the distributions better than DT, since CDT is aware of distributional information of entire trajectories.

| Method | halfcheetah | | | hopper | | | walker2d | | | Average |
|---|---|---|---|---|---|---|---|---|---|---|
| | Expert | Medium | Total | Expert | Medium | Total | Expert | Medium | Total | |
| Categorical DT | $1.270 \pm 0.242$ | $2.371 \pm 1.747$ | $1.821 \pm 1.363$ | $0.157 \pm 0.038$ | $0.337 \pm 0.088$ | $0.247 \pm 0.112$ | $0.289 \pm 0.052$ | $0.964 \pm 0.104$ | $0.626 \pm 0.347$ | 0.898 |
| DT | $1.173 \pm 0.372$ | $2.056 \pm 1.045$ | $1.614 \pm 0.901$ | $0.432 \pm 0.087$ | $0.531 \pm 0.053$ | $0.482 \pm 0.088$ | $0.408 \pm 0.246$ | $0.885 \pm 0.143$ | $0.646 \pm 0.312$ | 0.914 |

Table 15: Wasserstein-1 distance between shifted state-feature (x-velocity) distribution and the rollout distributions. Similar to the case of reward, Categorical DT handles the target distribution shifts and matches the distributions better than DT.

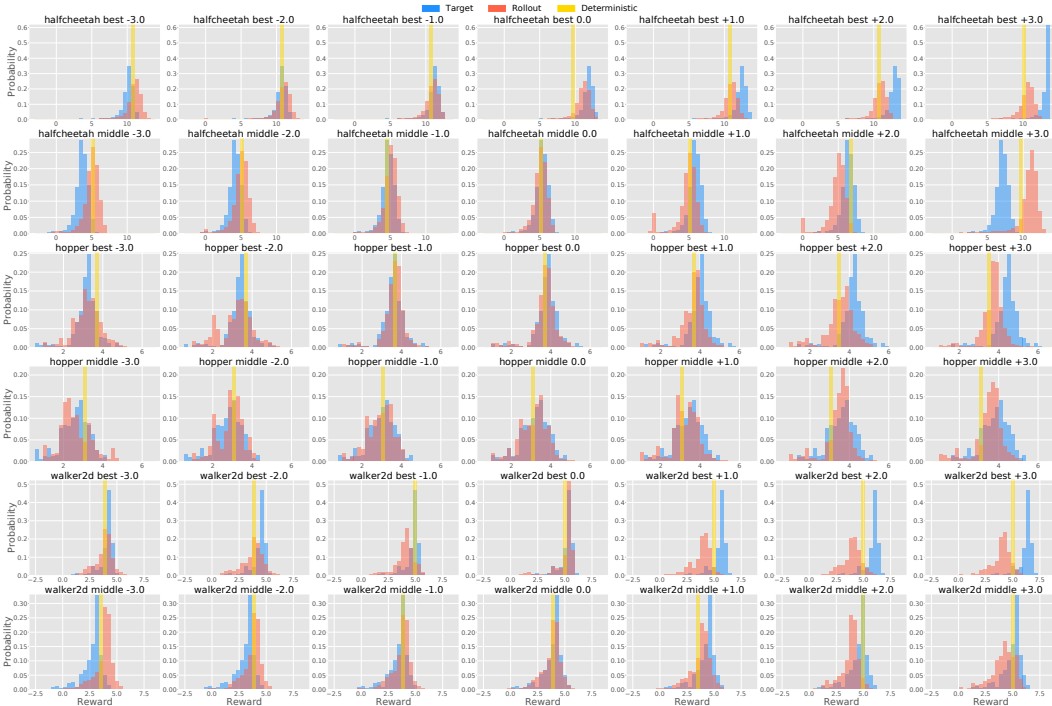

Figure 7: Reward distribution matching in halfcheetah (top two rows; best and middle), hopper (middle two rows; best and middle), and walker2d (bottom two rows; best and middle). We shift the target distribution with (from left to right column); `bin_size` $\times\{-3.0, -2.0, -1.0, 0.0, +1.0, +2.0, +3.0\}$ (Table 14). Categorical DT (red) can match the rollouts to the shifted target distributions (blue) when the shifted targets are within the support of dataset distributions. For DT (yellow, captioned as Deterministic), we only visualize the delta function at the means of rollouts.

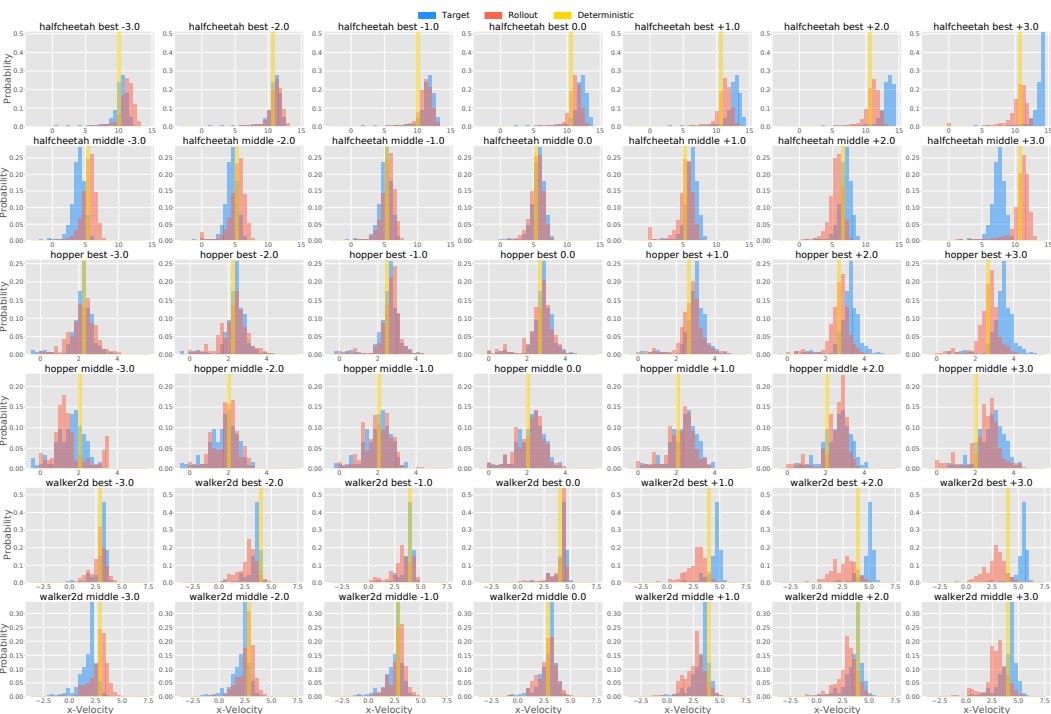

Figure 8: State-feature distribution matching, especially x-velocity, in halfcheetah (top two rows; best and middle), hopper (middle two rows; best and middle), and walker2d (bottom two rows; best and middle). We shift the target distribution with constant offset (from left to right column); bin_size $\times\{-3.0, -2.0, -1.0, 0.0, +1.0, +2.0, +3.0\}$ (Table 15). For DT (yellow, captioned as Deterministic), we only visualize the delta function at the means of rollouts.

### E.8 SYNTHESIZING UNREALISTIC TARGET DISTRIBUTION

We synthesize six target distributions manually generating x-velocity samples from Gaussian distributions via python scripts as done in prior works (Ghasemipour et al., 2020; Gu et al., 2021). Since we consider the one-dimensional feature space (x-velocity), we simply specify the mean and standard deviation of Gaussian distributions referring each dataset distribution, as shown in Figure 3, and then generate the 1000 samples per trajectory. While these targets are designed at least within the support of the dataset, we do not consider physical realizability.

- **HalfCheetah:** $(\mu, \sigma) = (5.0, 1.0), (13.0, 1.0), (9.0, 1.0), (2.5, 1.0), \{(5.0, 1.0), (13.0, 1.0)\}, \{(9.0, 1.0), (2.5, 1.0)\}$.
- **Hopper:** $(\mu, \sigma) = (2.5, 1.0), (1.5, 1.0), (3.5, 1.0), (2.5, 0.5), \{(3.5, 1.0), (1.5, 1.0)\}, \{(3.5, 0.5), (1.5, 0.5)\}$.
- **Walker2d:** $(\mu, \sigma) = (3.5, 1.0), (2.5, 1.0), (4.5, 1.0), (1.5, 1.0), \{(2.5, 0.5), (4.5, 0.5)\}, \{(1.5, 0.5), (4.5, 0.5)\}$.

For the last two sets, that have different distributional parameters $\{(\mu_1, \sigma_1), (\mu_2, \sigma_2)\}$, we sampled from two gaussian distributions 500 samples each, and marge them as one trajectory that has multiple modes.

Although they might be unrealistic, Figure 9 implies that CDT tends to match the target distributions, even in the cases of bi-modal target distributions. Such generalization to synthesized distribution is an important benefit of distribution-conditioned training. We also quantify the performance of CDT against DT from distributional matching perspective in Table 16.

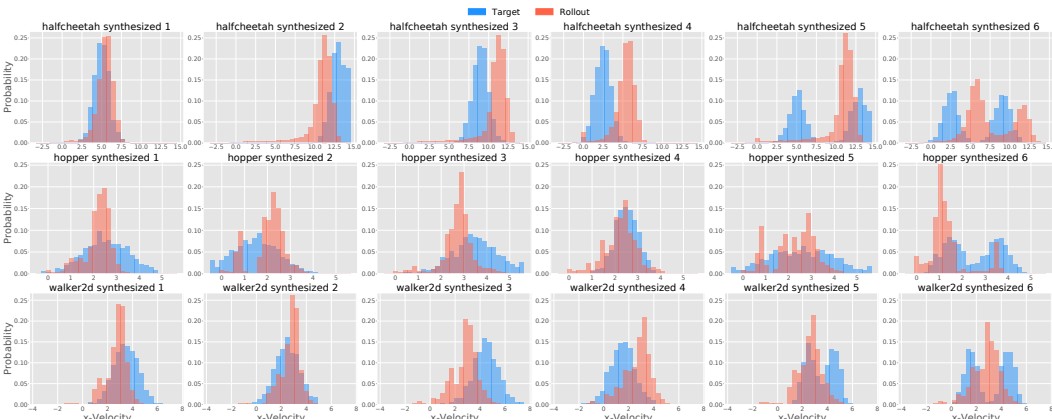

Figure 9: State-feature distribution matching in halfcheetah (top), hopper (middle), and walker2d (bottom). We synthesize the each target distribution from Gaussian distributions. While the results are worse than for realistic test target distributions in Figure 4, considering that many of these arbitrary synthetic targets could be unrealizable, seeing bi-modal matching results show that indeed CDT has learned to generalize something non-trivial.

| Method | halfcheetah | hopper | walker2d | Average |
|---|---|---|---|---|
| Categorical DT | $2.059 \pm 0.772$ | $0.536 \pm 0.225$ | $0.920 \pm 0.428$ | 1.172 |
| DT | $4.256 \pm 2.220$ | $0.584 \pm 0.298$ | $0.974 \pm 0.540$ | 1.938 |

Table 16: State-feature (x-velocity) distribution matching with synthesized, physically unrealistic target distributions generated from scripted Gaussian distributions. We compare Categorical DT and DT. Categorical DT manages to deal with such unrealistic target matching.

## F    DETAILS FOR CATEGORICAL DECISION TRANSFORMER

Categorical Decision Transformer (CDT) takes histograms of categorical distribution (i.e. discrete approximations of feature distributions; $B$-dim vector) as the inputs of the transformer. Here we describe how to compute the distributions for all timesteps given a trajectory $t \in [0, 1, \ldots, T]$. For simplicity, we explain the case of one-dimensional feature (e.g. scalar reward), but for $n$-dimensional features, we can adopt following procedure per each dimension and arrive at $n$ categorical features. This essentially approximates the full joints with a product of independent marginal distribution per dimension, and ensures that number of samples for getting reasonably discretized approximations do not need to grow exponentially as the dimension grows.

At first, we discretize the feature space $F$ using $B$ bins (per dimension), and convert the feature $\phi_t$ into the one-hot representation $\tilde{\phi}_t$. The range of feature space $[\phi_{\min}, \phi_{\max}]$ is pre-defined from the given offline datasets. $z_{\Phi_{\text{hist}}}(t)$, the categorical feature distribution at time step $t$, can be computed recursively following Bellman-like equation:

$$z_{\Phi_{\text{hist}}}(t) \propto \tilde{\phi}_t + \gamma(1 - \mathbb{1}[t = T])z_{\Phi_{\text{hist}}}(t+1), \tag{7}$$

where $\mathbb{1}$ is the indicator function. We compute a series of $z_{\Phi_{\text{hist}}}$ in a backward manner starting from $T$. After we obtain the desired information statistics for all trajectories, we feed them to the categorical transformer during training/test time. We describe the python-like pseudocode in Algorithm 1, coloring the changes from the original Decision Transformer (Chen et al., 2021a).

## G    DETAILS OF DECISION TRANSFORMER WITH LEARNED $\Phi$

While any unsupervised regularizer for an encoder could be combined into the action MSE loss of Decision Transformer to obtain the learned $\Phi$ efficiently, we observe that even simple objectives, such as auto-encoder and contrastive loss, sometimes perform well. For auto-encoder regularization (DT-AE), we train the MLP encoder (parameterized by $\psi$) and decoder with MSE loss of current state reconstruction:

$$\min_{\psi} \mathbb{E}_{s \sim D} \left[ \|s - \text{decoder}_{\psi}(\text{encoder}_{\psi}(s))\|^2 \right].$$

Then, we use the output of the encoder as a learned information statistics. In addition, for contrastive loss (DT-CPC), we adopt CURL objective (Srinivas et al., 2020) for state input while adding gaussian perturbation $\epsilon \sim \mathcal{N}(\mu = 0.0, \sigma = 0.1)$ as data argumentation (following Sinha et al. (2021)). We train the MLP encoder as a query, use its momentum encoder as a key:

$$\min_{\psi} \mathbb{E}_{\substack{s \sim D, \\ \epsilon, \epsilon' \sim \mathcal{N}(0, \sigma)}} \left[ \log \frac{\exp\left(\text{encoder}_{\psi}(s)^T W \text{encoder}_{\tilde{\psi}}(s + \epsilon)\right)}{\sum_{\epsilon' \neq \epsilon} \exp\left(\text{encoder}_{\psi}(s)^T W \text{encoder}_{\tilde{\psi}}(s + \epsilon')\right)} \right],$$

where $W$ is a learned parameter matrix for the bi-linear inner-product, and $\tilde{\psi}$ is the weights of the momentum encoder, updated as $\tilde{\psi} \leftarrow m\tilde{\psi} + (1 - m)\psi$, and $m = 0.95$. We treat its trained encoder output as a learned information statistics. It remains as future work to combine more advanced, temporary-extended objectives, such as attentive contrastive learning approach proposed in Yang & Nachum (2021). As described in Section 6.3, we consider three strategies to train the encoder for DT-AE and -CPC: training with only unsupervised loss, training with unsupervised and DT's supervised loss jointly (called as "joint"), and pre-training with only unsupervised loss and freezing the weights during DT training (called as "frozen"). We train DT-E2E with only DT's supervised loss as in Algorithm 1 for CDT and BDT.

**Algorithm 1 Categorical/Bi-directional Decision Transformer Pseudocode**: Orange and green texts describe additional details on top of the base DT pseudocode from Chen et al. (2021a).

```python
# z: information statistics (histogram, or learned representation)
# s, a, t: states, actions, or timesteps
# transformer: transformer with causal masking (GPT)
# embed_s, embed_a, embed_z: linear embedding layers
# anti_causal_tf: second transformer as encoder and aggregator
# embed_t: learned episode positional embedding
# pred_a: linear action prediction layer
# compute_stats: a function to compute information statistics

# main model
def DecisionTransformer(z, s, a, t):
    # compute embeddings for tokens
    pos_embedding = embed_t(t)
    s_embedding = embed_s(s) + pos_embedding
    a_embedding = embed_a(a) + pos_embedding
    if categorical:
        z_embedding = embed_z(z) + pos_embedding
    elif bi_directional:
        # input state sequence in a reverse order
        # NOTE: z is a target state sequence here
        reversed = flip(z)
        z_embedding = embed_z(anti_causal_tf(reversed)) + pos_embedding

    input_embeds = stack(z_embedding, s_embedding, a_embedding)

    # use transformer to get hidden states
    hidden_states = transformer(input_embeds=input_embeds)

    # select hidden states for action prediction tokens
    a_hidden = unstack(hidden_states).actions

    # predict action
    return pred_a(a_hidden)

# training loop
train_z = compute_stats(train_dataset)
# dims: (batch_size, K, dim)
for z, (s, a, t) in zip(train_z, train_dataset):
    if unsupervised:
        z = s
    a_preds = DecisionTransformer(z, s, a, t)
    loss = mean((a_preds - a)**2)
    optimizer.zero_grad(); loss.backward(); optimizer.step()

# evaluation loop
test_z = compute_stats(test_dataset)
n_tests = len(test_dataset)
for index in range(n_tests):
    test_trajectory = test_dataset[index]
    max_time_steps = len(test_trajectory)
    s, a, t, done = [env.reset()], [], [1], False
    # conditioning on the desired information statistics
    if categorical:
        z = [test_z[index][0]]
    elif bi_directional:
        z = [test_trajectory['observations'][0]]
    for t in range(max_time_steps):
        action = DecisionTransformer(z, s, a, t)[-1]
        new_s, r, done, _ = env.step(action)
        # append new tokens to sequence
        if categorical:
            z = z + [test_z[index][t+1]]
        elif bi_directional:
            z = z + [test_trajectory['observations'][t+1]]
        s, a, t = s + [new_s], a + [action], t + [len(z)]
        z, s, a, t = z[-N:], ...  # only keep context length of N
```

