# OpenReview forum: "Generalized Decision Transformer for Offline Hindsight Information Matching"
_ICLR.cc/2022/Conference — ICLR 2022 Spotlight_

### Official Review · Reviewer_Ffah · 2021-10-31

**Correctness:** 3
**Technical Novelty And Significance:** 3
**Empirical Novelty And Significance:** 3
**Recommendation:** 8
**Confidence:** 4

**Main Review:**

Strength:
a) unifying framework is developed and prior methods are nicely shown to be special cases
b) distributional and unsupervised versions of decision transformer are proposed
c) generalization for 1D distribution shifts is evaluated (i.e., distribution over x-component of velocity is shifted)

Weaknesses:
1) The paper is somewhat chaotic. It proposes 2 different extensions of decision transformer (DDT and UDT), furthermore introducing two variants of DDT (Categorical (CDT) and Gaussian (GDT)) and two variants of UDT (auto-encoder (AE) and contrastive loss) together with 4 settings (SL+UL, SL, UL, UL-Fix), at the same time it describes offline state-marginal matching and inverse-RL imitation learning and develops a hindsight information matching (HIM) framework. It feels there are enough ideas for several papers. Perhaps because of the scope, the evaluation is rather sketchy. Essentiall, the experiments only show that the basic implementation works, and that generalization w.r.t. to 1D velocity change seems to perform as expected. Focusing on just one variant, e.g., CDT, but providing a more thorough evaluation would be more beneficial and provide a clearer focus for the paper.
2) Gaussian Decision Transformer is announced but not described. Introduction refers to Sec. 5, which in turn refers to Appendix D, but the latter only covers Categorical Decision Transformer.
3) The way in which bold numbers are highlighted in tables may be misleading. In the majority of cases, results for DT and DDT lie within 1 standard deviation from each other. It would be more appropriate to either highlight both in bold or not highlight anything.
4) Only comparison to DT is provided. Comparison to baseline distributional algorithms should be added.

**Summary Of The Paper:**

The paper proposes an extension of Decision Transformer (DT) that can work with distributions of features. A number of prior hindsight-based or context-dependent methods are shown to be special cases of a generic scheme called Hindsight Information Matching (HIM), where arbitrary statistics of future trajectories can be used for conditioning. Therefore, the proposed Distributional Decision Transformer (DDT) can be seen as a practical implementation of the HIM algorithm. Experiments on D4RL medium-expert data for HalfCheetah, Hopper, and Walker2D investigate generalization of DDT. Additionally, another extension of DT called Unsupervised Decision Transformer (UDT) is proposed and evaluated that learns the features and rewards in unsupervised manner.

**Summary Of The Review:**

The paper is not ready for publication. The unifying perspective presented in the paper is very interesting, as well as the proposed distributional and unsupervised extensions of decision transformer. However, the evaluation is not sufficient. There are too many variations of the algorithm considered and the experiments are not expressive enough to illuminate them. Generalization only studied in a very limited scenario. No comparison to distributional RL is provided. As a result, the paper is trying to cover too many things at once and it is recommended to restrict the scope and provide more thorough analysis.

====

After rebuttal

The authors addressed my concerns with evaluations and baselines. Furthermore, the connection to distributional RL was clarified. The updated paper is 80% new. With the improvements made by the authors, I have no further issues and raise my score to "8 accept, good paper".

---

> ### Author Response · Authors · 2021-11-17
> **Response to Reviewer Ffah**
>
> We thank the reviewer for the careful reading and constructive discussions.
>
> **> Weaknesses 1**
>
> We modified Figure 1 to clarify the connection among our proposed methods.
> We introduce Generalized Decision Transformer framework and show how it leads to different classes of algorithms **with only small architectural changes**.
> For example, DT for offline RL can be recovered with $\Phi(s,a)=r(s,a)$ and $\gamma$-discounted summation as an anti-causal aggregator. We can get Categorical DT (CDT) for offline multi-task state-marginal matching when the aggregator is binning, and also get Bi-directional DT (BDT) for offline multi-task imitation learning if the aggregator is a second transformer (we empirically demonstrated that BDT is superior to other learned $\Phi$ approaches (DT-$X$, Section5.3), presented as unsupervised DT in the previous manuscripts).
> The choices of $\Phi(s, a)$ and the aggregator together decide $I^{\Phi}(\tau)$ in the Hindsight Information Matching (HIM) objective discussed in Section 4 and Table 1.
> We note that state-marginal matching (SMM) is one of the HIM problems, and the equivalence between SMM and Inverse RL is shown in previous work (Ghasemipour et al., 2020). We introduced all of these concepts from a unified perspective.
>
> We also strengthened the evaluation part; showing generalization of CDT to (1) 2D state-feature space (xy-velocities in Ant-v3; Appendix E.3), (2) different dimension (z-velocity) and synthesized bi-modal distribution (Section 6.2.1, the video is available at https://sites.google.com/view/generalizeddt), and (3) unseen velocity distributions during training (Section 6.2.2). Compared to baselines (DT, Meta-BC, and FOCAL), CDT shows strong distribution matching abilities and superior or comparable results across the experiments beyond simple 1D distribution matching. We also demonstrate that BDT outperforms the DT-$X$ variants, the learned $\Phi$ with unsupervised regularization (AE or CPC; see Section 5.2 and Appendix G), on offline one-shot distribution matching in full state (Section 6.3). The best results of BDT ($m=16, N=50$ in Table 5) outperform non-contextual BC, Meta-BC, and FOCAL baselines and is comparable to CDT or DT (in Table 2) even without manual state-feature specification.
>
>
> **> Weaknesses 2**
>
> We fixed this inconsistency by removing the description of Gaussian DT from the entire manuscript. In general, Gaussian DT may be interpreted as Generalized DT variants using different types of anti-causal aggregator, but we found that it in practice doesn’t work well because the statistics of feature $\Phi$ in the trajectories often don’t follow Gaussian distributions. Gaussian aggregators can’t capture an asymmetric nature, while the binning aggregators can.
>
>
> **> Weakness 3**
>
> In this revision, we removed the bold number from the table to prevent misleading and only highlighted the best average among the new “average” columns using a light blue mask.
>
>
> **> Weakness 4**
>
> First, we note that distribution matching (especially state-marginal matching) and distributional RL focus on different problems. While distribution or information matching algorithms try to solve Eq. 2 and Eq. 4, distributional RL solves the reward maximization problem (Eq. 1). Categorical DT has been inspired by Categorical Q Learning and adapted the discretization and binning, but it is not the distributional RL methods, rather a state-marginal matching method (we clarified this in Section 2, “Distributional Reinforcement Learning and State-Marginal Matching” in the revised paper).
> Due to these differences, offline distributional RL may not solve the distribution matching. In addition, previous SMM methods only work online because they require on-policy samples. Instead, we compared CDT to additional two baselines; Meta-BC and FOCAL (Li et al., 2021), a metric-based offline meta RL algorithm, because our problem settings resemble meta learning (see Appendix D for the details of those connections). Throughout the experiments, these methods provided good baseline performances to compare.
>
>
> **Summary**
>
> We hope the additional experiments to verify the generalization and introduction of the GDT framework can address all of your concerns around the evaluation and relation among the proposed methods (CDT, BDT, and DT-$X$). If the reviewer still has any remaining questions or concerns, we would be glad to hear that.
>
>
> **Reference**
>
> [Ghasemipour et al., 2020] A divergence minimization perspective on imitation learning methods. Conference on Robot Learning, 2020.
>
> [Li et al., 2021] FOCAL: Efficient fully-offline meta-reinforcement learning via distance metric learning and behavior regularization.  International Conference on Learning Representations, 2021.

---

### Official Review · Reviewer_5xwG · 2021-11-02

**Correctness:** 3
**Technical Novelty And Significance:** 3
**Empirical Novelty And Significance:** 3
**Recommendation:** 8
**Confidence:** 4

**Main Review:**

Strengths:
1. Novel work tackling offline state-marginal-matching (SMM) problem. Like other works learning RL skills from offline datasets, this can learn from more signal from fixed data than reward alone. From a learning standpoint, this can aid in learning much more interesting models, especially in the consideration of zero-shot or few-shot learning like GPT or BERT.
2. By conditioning on the state distribution of a new trajectory, DDT can serve as an imitator at test-time, similar to how language models/GPT can imitate prompts.
3. Similar to (2) but slightly distinct, this work extends DT to consider a distribution of rewards/returns, which is also novel itself.
4. Results show fairly strong conditioning performance across the settings considered; in particular I think Section 6.2.2 with an unrealistic target distribution and Section 6.3 with Unsupervised DT are interesting.
5. Proposed Hindsight Information Matching framework is an interesting way to interpret past work. While I'm not sure about the novelty exactly (i.e. log-probability, squared-error loss are well-known to connect to KL divergence), the work discusses many prior algorithms.
6. Investigates various representation learning approaches in the context of DTs.

Weaknesses:
1. While the promise of state-marginal-matching and language modeling is to extract a lot of signal from a dataset, namely trajectory statistics here, much of the experimental analysis focuses on (a) learning from a single feature (as far as I can tell) and (b) x-velocity in particular. On many of these environments, the reward is actually just the x-velocity shifted by some small amount, so it is almost identical to conditioning on the reward. Unsupervised DT lifts the restriction of (a) but is still evaluated on (b). A simple addition would be to evaluate on height; a more substantial addition would be different environments, see (4).
2. The proposed method seems very similar to one-shot imitation learning (OSIL) [1], where a model is trained to imitate a new trajectory at test time end-to-end, bypassing the statistics step. [2] tackles OSIL with transformers. While DDT/UDT may be more interpretable and/or powerful than OSIL, this is not compared to or discussed in the text.
3. In the contributions, it is written that the work "proposed the first benchmark for offline state-marginal matching". Table 2 is the only quantitative result shown in the paper, however the results seem hard to interpret, and I'm not sure the Wasserstein distance is the right choice of metric here. A proper quantitative metric is important for a benchmark. Adding qualitative comparisons to the main text for better clarity could be helpful. I am also not sure exactly how Deterministic DT works (what is it conditioned on?), but it is probably also a weak baseline. If conditioning on the reward is equivalent to the x-velocity, then Deterministic DT would be a strong baseline. Otherwise, other approaches like OSIL or conditioning on the mean of the target feature would be better.
4. I think the various representation learning methods investigated for Unsupervised DT are interesting, but they are not really shown or discussed in the main text, so it is hard to think of them as a main investigation of the paper. If the insights can be distilled into a paragraph/table/figure in the main text, I think it would be helpful.
5. In the introduction, it is written that UDT can "imitate any test trajectory". I would strongly recommend rewording this claim, as it is practically impossible to evaluate this claim in continuous settings.
6. It is not clear to me that the test-time target distributions proposed in Section 6.2 are interesting problems to study and especially to benchmark on. Manipulation tasks, or Atari and other rich environments, could generate more interesting heldout trajectories which are plausible and interesting, but differing in the statistics compared to the training set.

Overall, I am optimistic about this work, and interested in how we can maximally extract training signal from fixed datasets for reinforcement learning. In particular, I think a model which can effectively perform SMM may yield strong finetuning behaviors, although this could be left for future work. Learning to perform SMM offline may also aid in downstream online exploration, which is an important research direction. While I think the experiments and discussion are somewhat weak, I think some relatively simple adjustments (1-5 in weaknesses above) would improve the paper; if they are added, I will change my score to a weak accept. If the authors are interested in continuing this work, I think (6) would make the paper significantly stronger.

=======

Miscellaneous

1. I wasn't fully sure what "hindsight BC algorithms always work with samples $(\tau_i, z^{\tau_i}$" was supposed to mean; I think hindsight RL methods do this as well? I might be misunderstanding something.
2. The tables in general have a small font which makes them hard to read, especially Tables 2, 8, and 9 are overloaded. It would be much easier to read with less decimal places or multiplied by 100 or some other choice; an aggregated mean in a single column would also make it easier to interpret. The figure font is also a little small at times.
3. Under Equation 1 in the definition of $p_t^\pi(s)$, the $t$ in the product should be redefined to some other $t'$ to not clash with the $t$ representing the time of marginal distribution.
4. In Equation 5, I think the expectation is missing $s \sim p_z^\pi$. (It is included in Equations 4 and 6, for instance)
5. For the pseudocode in the appendix, I would strongly recommend avoiding the use of red/green due to colorblindness issues.

=======

[1] Yan Duan et al. 2017. "One-Shot Imitation Learning"

[2] Sudeep Dasari and Abhinav Gupta 2020. "Transformers for One-Shot Visual Imitation"

**Summary Of The Paper:**

This work discusses many prior methods under a Hindsight Information Matching (HIM) framework, where the methods can be interpreted as  trying to minimizing the KL divergence between the achieved statistic and some target statistic for a particular choice of statistic. The authors propose to replace the return-to-go conditioning in Decision Transformer with a distribution over some specified state features, which they name Distributional Decision Transformer (DDT). This work also proposes an unsupervised variant, Unsupervised DT (UDT), which does not need certain state features to be specified and can be interpreted as performing offline state-marginal-matching. The authors show in various MuJoCo settings (HalfCheetah, Hopper, Walker2d) that the proposed methods can learn to imitate target information statistics in offline RL and imitation learning settings.

**Summary Of The Review:**

The paper is interesting and presents novel methods which may in the future be important ideas leading to new work. Since the results are somewhat weak right now and there are minor clarity concerns, I recommend weak reject.

=====

After rebuttal (11/18):

I think the updates made by the authors during the rebuttal period are significant, adding writing clarity, experiments, and the Bidirectional DT. As I wrote below in my response: "While the scope of experiments is still limited (as also pointed out by Reviewer Ffah), I think the experiments are thorough and the paper contains interesting insights that will be valuable for future investigation, especially given the early state of literature on transformers for RL; I will thus increase my score from 5 to 8."

---

> ### Author Response · Authors · 2021-11-17
> **Response to Reviewer 5xwG (1/2)**
>
> We appreciate the reviewer for the careful reading of our paper and detailed discussions.
>
> **> Weakness 1**
>
> In the revised paper, we added the 2D state-feature matching experiment (xy-velocities of
> Ant, Section E.3, Table 11), and z-velocity HalfCheetah (Section 6.3) to show the scalability of Categorical DT (CDT) to multi-dimensional or other state-feature settings.
>
> For 2D experiments in Appendix E.3, CDT matches the target distribution better than other baselines (DT, Meta-BC, FOCAL (Li et al., 2021)). This result indicates that CDT could handle multi-dimensional settings beyond 1D state-feature/reward matching.
>
> To generate more diverse behaviors in the z-axis, we obtained the expert cheetah that backflips towards -x direction by modifying reward function, and constructed the novel diverse behavior dataset combining expert backflipping trajectories and expert running forward trajectories from D4RL dataset (we described the data generation in Appendix E.4).
> We evaluated CDT, DT, Meta-BC, and FOCAL with not only uni-modal target trajectories (just running forward, or backflipping in a single rollout), but also synthesized bi-modal trajectories not provided during training (concatenating half of running forward trajectories to half of backflipping trajectories forcedly).
> Table 3 shows that CDT matches the distribution better than DT or FOCAL, and is competitive to Meta-BC that is originally designed to deal with such multi-task data.
> Figure 2 (a) also shows that CDT successfully matches the distribution to both uni- and bi-modal behaviors, while DT tends to lean backflipping behaviors and fails to fit them.
> CDT obtains the patchworked bi-modal behaviors, running forward first, then backflipping during a single rollout, successfully. The video of the learned bi-modal behavior is available at https://sites.google.com/view/generalizeddt.
>
> **> Weakness 2**
>
> As the reviewer pointed out, one-shot imitation learning (IL) can be interpreted as a part of hindsight information matching (HIM) algorithms, with the identity function as $\Phi$. We updated Table 1, including Duan et al., 2017, and added the discussion about the similarity between meta RL/IL and our settings in Appendix D. Also, we included Meta-BC and FOCAL, a metric-based offline meta RL method, as a baseline of the offline multi-task state-marginal matching problem (See Appendix B for the details of baseline).
>
>
> **> Weakness 3**
>
> First, we would like to mention that previous studies on state-marginal matching (Lee et al., 2020; Ghasemipour et al., 2020; Gu et al., 2021) studies have suffered from the lack of quantitative evaluations, because both state-marginal and target distribution are non-parametric; we cannot access their densities, and only their samples are available.
> Since the discretization of low-dimensional features, such as reward, have success in many RL methods (Bellemare et al., 2017; Dabney et al., 2018), quantification of distribution matching performance by Wasserstein-1 distance could be reliable evaluations (in addition Wasserstein-1 distance is symmetric, different from KL distance that is asymmetric). See Section 5.1 and Appendix C for the details.
>
> Deterministic DT is the same as Decision Transformer. We fixed the name on the tables.
>
> In the revised paper, we included 5 benchmark tasks (reward or x-velocity distribution matching (Section 6.1), xy-velocities distribution matching (Appendix E.3), z-velocity distribution matching with synthesizing bi-modal behaviors (Section 6.2.1), diverse x-velocity distribution matching with unknown targets (Section 6.2.2), and reward or x-velocity distribution matching in full state (Section 6.3)) with quantitative evaluations (Table 2-5 + Appendix E) and some qualitative evaluations (Figure 2 + video of bi-modal behaviors), compared to DT, Meta-BC, and FOCAL.
>
> **> Weakness 4 and 5**
>
> In the revised paper, we appropriately toned down the claim for the learned $\Phi$ methods (DT-$X$ and BDT), such as *We also demonstrate the effectiveness of Bi-directional DT as a one-shot imitation learner, significantly outperforming simple variants based on DT.* in Conclusion.
> Through the experimental evaluation, we found that just adding auto-encoder (AE) or contrastive (CPC) regularizer is not sufficient enough to learn $\Phi$ without manual feature specification, and in the additional experiments, we also found that the aggregator choice (in Figure 1) plays an important role to extract sufficient information for control.

---

> > ### Author Response · Authors · 2021-11-17
> > **Response to Reviewer 5xwG (2/2)**
> >
> > **> Weakness 6**
> >
> > As mentioned above (response to Weakness 1), we added the variety of experiments with (1) Ant (for 2D case), (2) backflipping cheetah (for z-velocity case), and (3) unseen cheetah velocity tasks (as meta-learning-inspired tasks with diversified datasets), and moved the experiments of constant shifting and synthesizing unrealistic target distribution to Appendix E (these settings are studied in previous works (Ghasemipour et al., 2020; Gu et al., 2021)).
> > Especially, (2) backflipping cheetah with synthesized bi-modal behavior (running forward first, and then backflipping during single rollout), and (3) unseen cheetah velocity tasks held-out several target velocities successfully demonstrated the generalization of CDT when given diverse datasets.
> > We agree that studying manipulator or point mass tasks (as in  Ghasemipour et al., 2020) and image-based tasks such as Atari are interesting directions for future work.
> >
> >
> > **> Miscellaneous 1**
> >
> > We made a clarification at this point in Section 4, third bullet point: as shown in Table 1, only goal-based or multi-task $I^{\Phi}(\tau)$ can use RL methods to solve HIM problem, while all four plus our distribution-based $I^{\Phi}(\tau)$ can use BC.
> >
> >
> > **> Miscellaneous 2**
> >
> > We thank the reviewer for raising this issue. We resolved the issue of small order in the evaluation metric by fixing the bugs contained in the evaluation code and by leveraging reliable python libraries to compute Wasserstein-1 distance: https://github.com/pkomiske/Wasserstein. In addition, we added “Average” columns to improve the visibility.
> >
> > **> Miscellaneous 3 & 4**
> >
> > We fixed the notations in this revision accordingly.
> >
> >
> > **> Miscellaneous 5**
> >
> > We also thank the reviewer for pointing out the coloring issue. While we adjusted Algorithm 1 in Appendix F to the revised paper, we also changed the color map to the colorblindness-friendly ones. We referred https://personal.sron.nl/~pault/ as a reference. In addition, the light blue mask in the table also follows this coloring.
> >
> > **Summary**
> >
> > We hope our response can address all of your concerns raised in the rebuttal. If the reviewer still has any remaining questions or concerns, we would be glad to hear that.
> >
> >
> > **Reference**
> >
> > [Li et al., 2021] FOCAL: Efficient fully-offline meta-reinforcement learning via distance metric learning and behavior regularization. International Conference on Learning Representations, 2021.
> >
> > [Duan et al., 2017] One-Shot Imitation Learning. Advances in Neural Information Processing Systems, 2017.
> >
> > [Lee et al., 2020] Efficient exploration via state marginal matching.arXiv preprint arXiv:1906.05274, 2020.
> >
> > [Ghasemipour et al., 2020] A divergence minimization perspective on imitation learning methods. Conference on Robot Learning, 2020.
> >
> > [Gu et al., 2021] Braxlines: Fast and interactive toolkit for rl-driven behavior engineering beyond reward maximization.arXiv preprint arXiv:2110.04686, 2021.
> >
> > [Bellemare et al., 2017] A distributional perspective on reinforcement learning. International Conference on Machine Learning, 2017.
> >
> > [Dabney et al., 2018] Distributional reinforcement learning with quantile regression. Thirty-Second AAAI Conference on Artificial Intelligence, 2018.

---

> > ### Comment · Reviewer_5xwG · 2021-11-18
> > **Response to rebuttal**
> >
> > Thanks for the updates! I think the updates to the writing are very helpful, and also that the newly proposed Bidirectional DT is also quite clever and can be related to prior work: (1) whereas OSIL requires training trajectories to be categorized by task/goal, BDT utilizes the HIM idea applied to the trajectory and can operate on miscellaneous trajectories; and (2) rather than using a backwards context as in (Rakelly et al. 2019; Fakoor et al. 2020), a forwards context is utilized to specify desired behavior. Again a minor weakness is that BDT is not evaluated on tasks which are considered interesting in OSIL literature, but the paper has enough other contributions that I think it is fine.
> >
> > One point I think is still remaining is that I don't think the details for Deterministic DT are written in the main text; I could only find in Section E.3 that Deterministic DT is given the mean x- or y-velocity (I assume analogously set up for the other experiments), which is I think still nontrivial from the original DT formulation.
> >
> > I also missed what the datasets used in the Ant experiments are. Ant-v3 is notoriously hard in the D4RL setting (i.e. non-expert trajectories), so many papers don't evaluate on it, and thus if the data is non-expert and good performance is learned (possibly with qualitative visualizations), then this would also be very interesting.
> >
> > Seeing the new results in Table 5 comparing the various representation learning methods, I am curious as to why AE outperforms CPC consistently (except in the case of joint supervised training). Is there any intuition or guidance as to why this happens?
> >
> > While the scope of experiments is still limited (as also pointed out by Reviewer Ffah), I think the experiments are thorough and the paper contains interesting insights that will be valuable for future investigation, especially given the early state of literature on transformers for RL; I will thus increase my score from 5 to 8.
> >
> > =====
> >
> > [1] Kate Rakelly et al. 2019. "Efficient off-policy meta reinforcement learning via probabilistic context variables"
> >
> > [2] Rasool Fakoor et al. 2020. "Meta-Q-Learning"

---

> > > ### Author Response · Authors · 2021-11-19
> > > **Followup Comments**
> > >
> > > We thank the reviewer for the quick response to our update. We updated the manuscript again reflecting your additional feedback as summarized below.
> > >
> > > **> Details of Decision Transformer**
> > >
> > > We added the description of the procedure for original DT in state-feature (e.g. x- or z-velocity) settings.
> > >
> > > - “As the same as the reward case, DT takes the summation of the state-feature over a trajectory as an input.” (in Section 6.1)
> > > - “For DT, we compute the summation of x- and y-velocity each over trajectories and normalize them with the maximum horizon. DT feeds these two scalars as information statistics to match.” (in Appendix E.3)
> > >
> > > **> Details of 2D state-feature distribution matching**
> > >
> > > Same as the other environments (HalfCheetah, Hopper, Walker2d), We used the medium-expert-v2 dataset for the experiments of Ant (in Appendix E.3). We also added the visualization of x- and y-velocity distributions respectively (Figure 5 in Appendix E.3), which shows that while CDT could cope with the distribution shift between expert and medium target distribution, DT always fits the medium one even if the expert trajectory is given as a target. Interestingly, the CDT's improvement of distribution matching over DT in Ant seems more substantial than in other environments (compared to Table 2). These observations imply that CDT could extract more learning signals efficiently than DT.
> > >
> > >
> > > **> DT-AE vs DT-CPC**
> > >
> > > DT-AE vs DT-CPC comparison is an interesting question. While we cannot provide definite answers, we speculate some of the following: (1)  AE can utilize more learning signals since it has a reconstruction objective, (2) our state space is low-dimensional, so it's still tractable for AE to learn good representations (e.g. unlike in many image domains where CPC may provide better features [van den Oord et al., 2018; Srinivas et al., 2020]), (3) in Dreamer [Hafner et al., 2020], it was also shown that AE approach (reconstruction) outperforms CPC in most environments.
> > >
> > > **> Minor Update**
> > >
> > > We included Fakoor et al., 2020 as a reference.
> > >
> > >
> > > **Reference**
> > >
> > > [Fakoor et al., 2020] Meta-Q-Learning. International Conference on Learning Representations, 2020.
> > >
> > > [Srinivas et al., 2020] CURL: Contrastive unsupervised representations for reinforcement learning. International Conference on Machine Learning, 2020.
> > >
> > > [van den Oord et al., 2018] Representation learning with contrastive predictive coding. arXiv preprint arXiv:1807.03748, 2018.
> > >
> > > [Hafner et al., 2020] Dream to Control: Learning Behaviors by Latent Imagination. International Conference on Learning Representations, 2020.

---

> > > > ### Comment · Reviewer_5xwG · 2021-11-19
> > > > **Response to followup**
> > > >
> > > > Thanks again for the updates!
> > > >
> > > > I agree the improvement in the Ant environment vs DT is interesting; if the writing in the main text could be made more concise, it would be interesting to add the results to the main text.
> > > >
> > > > Thanks also for the pointer to the Dreamer results with AE/CPC; I missed this previously.

---

### Official Review · Reviewer_cUy5 · 2021-11-13

**Correctness:** 3
**Technical Novelty And Significance:** 2
**Empirical Novelty And Significance:** 2
**Recommendation:** 6
**Confidence:** 4

**Main Review:**

This paper offers three main contributions:
1. It consolidates a number of existing techniques, including TDMs, Learning from Play, GCSL, and Decision Transformer (DT), into the same umbrella of "hindsight information matching" algorithms in the spirit of hindsight relabeling.
2. It describes a variant of DT, called Categorical DT, that conditions on a desired future state or feature distribution (as opposed to scalar cumulant).
3. It offers a version of the Categorical DT that includes an unsupervised learning component, with the goal being to learn representations $\phi$ such that matching the first moment of $\phi$ is equivalent to matching the distribution of raw features.

**Hindsight information matching**

I did appreciate the exercise of mapping out the space of hindsight algorithms that resulted in Table 1. I am not sure this includes a new insight, since hindsight is commonly acknowledged as a primary motivation for algorithms in this space. For example, from the Decision Transformer paper:
> Thus, by combining the tools of sequence modeling with **hindsight return information**

and from the Trajectory Transformer paper:
> ...our method also resembles recently proposed work on goal relabeling [**HER**, GCSL]
and reward-conditioning [UDRL, RCP] to reinterpret all past experience as useful demonstrations with
proper contextualization.

and from the GCSL paper:
> By generating demonstrations using **hindsight relabelling**, we are able to apply goal-conditioned imitation learning primitives...

That being said, I think it is valuable to aggregate trends like these in a way that makes it easier for researchers to understand how a number of related algorithms fit together, so even if there is nothing technically new from this exposition I do think it is a useful contribution.

**Categorical DT**

I was somewhat less convinced of the other two contributions.
While in principle the ability to match a target _distribution_ should enable substantially more flexible behavior than scalar return-conditioning, this did not seem to be adequately explored or evaluated.
One low-level issue is the datasets used for evaluation: all experiments use the `medium-expert` D4RL datasets from what I can tell, which are mixtures of only two policies.
This severely limits the types of behaviors that can be extracted from the datasets, and calls into question even some of the more impressive-looking generalization results.

For example, in Figure 4 we see that the model is able to produce a bimodal x-velocity distribution given a bimodal input (halfcheetah synthesized 6, top right).
Somewhat suspiciously, however, the modes are in the wrong places.
The reason the modes land where they do is because these are the modes of the two policies in the dataset: the medium policy gets an average of 5~6 reward per step, and the expert policy gets an average of ~12 reward per step.

While this is possibly the best that one could hope for given the dataset, it sets the bar fairly low.
Since there are only two behaviors in the dataset, it is only possible to recover different mixtures of two qualitatively different things.
You could reduce the entire distributional conditioning to a single parameter that acts as a mixing coefficient between the two policies.
In general, this sort of idea would be better evaluated in datasets with higher entropy, to better show the types of behaviors that are recoverable from distribution conditioning that would not be recoverable via standard return conditioning.

**Unsupervised DT**

Unfortunately I could not find quite enough detail about either the method or evaluation here to understand what was going on.
I think I understand how a representation learning objective could also be used for distribution matching purposes, but aside from that high-level link this addition did come off as a bit tacked-on to the end of another paper.
Most of the design choices are relegated to the appendix, which is sometimes reasonable given space constraints, but then even the appendix does not provide a self-contained description but mostly references out to other papers, such as:
> In addition, for contrastive loss, we adopt CURL objective (Srinivas et al., 2020) for state input while removing data augmentation.

More detail could be useful here -- the data augmentation is important to contrastive methods like CURL, so I am not entirely sure what objective is being used.

The results are somewhat terse as well:
> We found that generally SL loss with respect to $\phi$ could ruin representation learning, as results for SL + UL and SL demonstrate unsuccessful learning , and using contrastive loss is worse than using AE loss (see Figure 15 in Appendix).

If three of the four variants of Unsupervised DT are meant to serve as either ablations or negative results, could you write more about what we should learn from them?
I think it could be interesting that the unsupervised-only version works so much better than versions with the supervised loss, but without a more thorough investigation into why this happens it is hard to contextualize.

**Minor**
1. "policy that whose future state rollouts" --> "policy whose future..."
1. Footnote 7: "prameterized" --> "parameterized"
1. " To compute $I_z^\phi(\tau_t:T)$ efficiently for all timesteps $t$ given a trajectory $\tau_1:T$ , we use similar recursive Bellman-like computation inspired by Bellemare et al. (2017)." Could you write the recursion to make the description more self-contained?
1. "CDT don’t match the targets" --> "CDT doesn't..."
1. "unsuccessful learning ,": remove space before comma

**Post-rebuttal update**

Thanks to the authors for their responses. I have updated my score based on the conversation below.

**Summary Of The Paper:**

This paper presents variants of the Decision Transformer that can condition on desired state or feature distributions instead of scalars.
Along the way, the authors also describe an organization of hindsight relabeling methods proposed in the last few years.

**Summary Of The Review:**

This paper presents a great and potentially very useful idea, but does not evaluate it in situations where the idea could reasonably be expected to do something more interesting than the original DT.
Better motivating the types of behaviors that could be expressed and produced via more flexible conditioning, and seeing that through by testing the method on higher-entropy datasets, would make this a much stronger contribution.

---

> ### Author Response · Authors · 2021-11-17
> **Response to Reviewer cUy5 (1/2)**
>
> We thank the reviewer for the careful reading of our paper and constructive comments in detail.
>
>
> **> Hindsight Information Matching (HIM)**
>
> As the reviewer mentioned, our unification not only clarifies the difference or relationship among HIM algorithms, but also provides the following novel insights:
>
> - New HIM algorithms can be proposed by simply changing $I^{\Phi}(\tau)$, as we did to propose Categorical DT for distribution-based problems.
> - Given a choice of $I^{\Phi}(\tau)$, new HIM algorithms can be proposed by changing implementation details, such as using “RL" or “BC" as algorithm type, doing “Online” or “Offline” training, and network architectures.  All “Offline” “BC” methods could be adopted easily to “Online” learning through iterative data collections (Ghosh et al., 2021; Kumar et al., 2019).
> - Only (1) goal-based and (2) multi-task problem can use “RL” as Algorithm Type, while all four, plus our (5) distribution-based, can use “BC”, because “RL” requires optimizing Eq. 4 with respect to the policy, which gets non-trivial for some choices of $I^{\Phi}(\tau)$; e.g. (3)-(5) return-based, full trajectory imitation, or distribution-based problem. “BC” bypasses the need to solve Eq. 4 and therefore is universally applicable to any $I^{\Phi}(\tau)$ or HIM algorithm.
>
> The last insight motivated us to invent the “offline” and “BC” methods; GDT, to optimize information matching objectives. We updated Table 1 and described them explicitly in the latest version of the manuscript (Section 4) to clarify our contributions.
>
>
> **> Categorical DT**
>
> We added two novel offline distribution matching experiments with more diverse behavior (higher-entropy) datasets than D4RL medium-replay: (a) z-velocity matching with running forward or backflipping behaviors (Section 6.2.1), (b) target x-velocity matching unseen during training adapted from meta learning task (Section 6.2.2). Let us describe those details and results briefly.
>
> **(a) z-velocity matching with running forward or backflipping behaviors (Section 6.2.1)**
> To generate more diverse behaviors in the z-axis, we obtained the expert cheetah that backflips towards -x direction by modifying reward function, and constructed the novel diverse behavior dataset combining expert backflipping trajectories and expert running forward trajectories from D4RL dataset (we described the detailed data generation process in Appendix E.4).
> We evaluated CDT, DT, Meta-BC, and FOCAL (Li et al., 2021) with not only uni-modal target trajectories (just running forward, or backflipping in a single rollout), but also synthesized bi-modal trajectories not provided during training (concatenating half of running forward trajectories to half of backflipping trajectories forcedly).
> The results in Table 3 show that CDT matches the distribution better than DT or FOCAL, and is competitive to Meta-BC that is originally designed to handle such multi-task data.
> Figure 2 (a) also shows that CDT successfully matches the distribution to both uni- and bi-modal behaviors, while DT tends to lean backflipping behaviors and fails to fit them.
> CDT obtains the patchworked bi-modal behaviors, running forward first, then backflipping during a single rollout, successfully. The video of the learned bi-modal behavior is available at https://sites.google.com/view/generalizeddt.
>
> **(b) target x-velocity matching unseen during training adapted from meta learning task (Section 6.2.2)**
> Generalization to the unknown target demonstrations or tasks has been actively investigated in meta or one-shot RL/IL literature (Duan et al., 2016; Wang et al., 2016). So we adapted the popular task from the meta learning domain to our offline state-marginal matching problem: cheetah velocity tasks, where the agent tries to run with specified x-velocity.
> We prepare 31 target x-velocities; taken from [0.0, 3.0], uniformly spaced at 0.1 intervals, and hold out {0.5,1.5,2.5} trajectories as a test set for simplicity (we described the detailed data generation process in Appendix E.5).
> The results in Table 4 present that CDT outperforms DT or FOCAL, and is slightly better than Meta-BC, which implies CDT can generalize the unseen target trajectories to match given sufficiently diverse offline datasets. Figure 2 (b) also shows that CDT successfully handles the trajectories unseen during training, while DT seems to output covering behaviors over the dataset support.
>
> We included those two as main results and moved the experiments of constant shifting and synthesizing unrealistic target distribution to Appendix E. We hope our extra experiments with higher-entropy datasets address your concerns on the contribution of CDT.

---

> > ### Author Response · Authors · 2021-11-17
> > **Response to Reviewer cUy5 (2/2)**
> >
> > **> Unsupervised DT**
> >
> > Through the previous Unsupervised DT experiment, we found that just adding auto-encoder (AE) or contrastive (CPC) regularizer isn’t sufficient enough to learn $\Phi$ without manual feature specification, and in the additional experiments, we also found that the aggregator choice (in Figure 1) plays an important role. In the revised paper, we introduced Bi-directional Decision Transformer (BDT), which does not receive any explicit feature function $\Phi$ and rather obtains it through a parameterized aggregator, utilizing a second anti-causal transformer that takes a reverse-order state sequence as an input. We also renamed the previous “Unsupervised DT” as DT-AE,  DT-CPC, or DT-E2E (see Section 5.3) for the comparison.
> > In Section 6.3, we compare BDT, DT-AE, and DT-CPC as a one-shot distribution matching problem given full state. The results in Table 5 show that BDT outperforms all of other learned $\Phi$ variants or Meta-BC and is comparable to CDT or DT (in Table 2) with longer input (N= 50), while even simple approaches, DT-AE and DT-AE (frozen), show positive results compared to no-context BC baselines presented in Table 2. This implies that even though we don’t assume the state-feature specification, aggregator choice in GDT with minimal architectural changes may solve the offline distribution matching problem efficiently.
> >
> > As for DT-CPC, we use a CURL-like objective ​​adding Gaussian perturbation $N(\mu= 0.0, \sigma= 0.1)$ to the input state as data argumentation (following Sinha et al., 2021). We found that similar to the discussion in Yang & Nachum, 2021, DT-CPC sometimes fails to obtain sufficient representation for imitation, and we guess a temporary-extended objective might be required.
> >
> >
> > **> Minor Comments**
> >
> > We resolved all the comments the reviewer pointed out in the revised manuscript. For Minor 3, see Appendix F for the details of recursive Bellman-like computation.
> >
> > **Summary**
> >
> > We appreciate the reviewer again for the detailed review. If the reviewer still has any remaining questions or concerns, we would be glad to hear that.
> >
> >
> >
> > **Reference**
> >
> > [Ghosh et al., 2021] Learning to reach goals via iterated supervised learning.  InInternationalConference on Learning Representations, 2021.
> >
> > [Kumar et al., 2019] Reward-conditioned policies.arXiv preprint arXiv:1912.13465, 2019.
> >
> > [Li et al., 2021] FOCAL: Efficient fully-offline meta-reinforcement learning via distance metric learning and behavior regularization.  InInternational Conference on Learning Representations, 2021.
> >
> > [Duan et al., 2016]  RL2: Fast reinforcement learning via slow reinforcement learning. arXiv preprint arXiv:1611.02779, 2016.
> >
> > [Wang et al., 2016] Learning to reinforcement learn. arXivpreprint arXiv:1611.05763, 2016.
> >
> > [Sinha et al., 2021] S4RL: Surprisingly simple self-supervision for offline reinforcement learning.arXiv preprint arXiv:2103.06326, 2021.
> >
> > [Yang & Nachum, 2021] Representation matters: Offline pretraining for sequential decisionmaking. InInternational Conference on Machine Learning, 2021.

---

> > > ### Comment · Reviewer_cUy5 · 2021-11-23
> > > **Re: Response to Reviewer cUy5**
> > >
> > > Thanks for the thorough response and marking the changes in another color.
> > >
> > > A few questions about the updates:
> > >
> > > 1. What is the intuition for CDT outperforming DT in the held-out x-velocity tasks? In tasks for which the distributional conditioning contains more information than scalar conditioning, I can see why the behavior would be different, but isn't the conditioning a scalar here for both cases?
> > >
> > > 2. > However, [the Wasserstein distance] is often intractable to measure such loss for full state or even for some state
> > > dimensions analytically because both state-marginal and target distributions can be non-parametric
> > > and we cannot access their densities. In practice, we empirically estimate it employing the binning of
> > > several state dimensions we specified. More discussions are included in Appendix C.
> > >
> > > I didn't find the information about binning or how the Wasserstein is approximated (other than a link to the repo). Could you describe how the binning works and what the "specification" of dimensions here means?
> > >
> > > 3. What do you condition DT on for the bimodal z-velocity experiments in Figure 2? If it is something like the average z-velocity, what happens with DT if you condition on the average of one mode for half of the rollout, and then switch to conditioning on the average of the other mode?
> > >
> > > One of my previous criticisms was that the medium-expert datasets probably did not provide an interesting enough setting to see the benefits of distributional conditioning. While I think that probably is true for most mixture distributions of only two policies -- since most behaviors are probably going to reduce to imitating one mode or the other, reducing to conditioning on one binary variable -- the running-flipping cheetah experiments are definitely an improvement.
> > >
> > > 4. I am still having a slightly hard time following the UDT (and BDT) descriptions. I see you've added the data augmentation term in Appendix G, but is there an objective provided somewhere? This is the closest I can find:
> > > >We train the MLP encoder as a query, use its momentum encoder as a key, and treat its trained encoder output as a learned information statistics.
> > >
> > > I see that it also follows the protocol from S4RL, but if there is a succinct-but-more-precise description, it might make the paper more self-contained.

---

> > > > ### Author Response · Authors · 2021-11-23
> > > > **Followup Comments**
> > > >
> > > > We thank the reviewer for the quick and detailed response to our update. We updated the manuscript again answering your additional concerns:
> > > >
> > > > **> 1. The intuition for CDT outperforming DT**
> > > >
> > > > CDT takes **histograms** of categorical distribution of the feature $\phi$ (i.e. **B-dim vector**) as the inputs of the transformer (we clarified this in Appendix F of the latest version), while DT does scalar; summation of the feature $\phi$ over trajectories (divided by maximum timesteps). As the reviewer mentioned, CDT is aware of the distributional information for the matching much richer than DT, which might lead to improvement.
> > > >
> > > > **> 2. The detailed explanation of Wasserstein distance**
> > > >
> > > > We additionally described the detailed procedure to compute Wasserstein distance in the end of Appendix C.  We empirically estimate Eq. 5 by focusing on the
> > > > “manually-specified” features $\phi \in F$ (e.g. we chose reward or xyz-velocities in this work) and employing binning and discretization. We discretize the feature space $F$ and obtain $B$ bins (per dimension), whose representatives are $\bar{\phi}_{l} (l = 1, ..., B)$.
> > > > Then we get the histograms of the feature $\phi$ from the target trajectories and rollouts of the policy. Leveraging these values, we compute the evaluation metric as described in Eq. 6 (solving the constraint optimization problem with the python package: https://github.com/pkomiske/Wasserstein).
> > > >
> > > >
> > > > **> 3. The bimodal z-velocity experiments in Figure 2**
> > > >
> > > > Similar to the x-velocity experiments in Section 6.1 (Table 2), DT is conditioned on the summation of the feature $\phi$ over concatenated bi-modal trajectories (divided by maximum timesteps). While we don’t have the results in the exact same setting as the reviewer mentioned (“conditioning on the average of one mode for half of the rollout, and then switching to conditioning on the average of the other mode”), we speculate that the cheetah just does backflipping. This is because DT can't match the even uni-modal forward-running behavior (Figure 2 (a), the last row, the first column), and leans towards backflipping behaviors.
> > > >
> > > >
> > > > **> 4. UDT (DT-$X$ and BDT) descriptions**
> > > >
> > > > We added the explanation of UDT in Appendix G, describing the objective of DT-CPC explicitly.
> > > > UDT (DT-$X$) optimizes the combination between DT's objective (MSE of the action; described in Algorithm 1) and AE/CPC objectives, with three different strategies. BDT optimizes MSE of the action the same as CDT, DT, and DT-E2E.
> > > >
> > > > If there are still unclear points or insufficient descriptions to the reviewer, please let us know freely.

---

> > > > > ### Comment · Reviewer_cUy5 · 2021-11-25
> > > > > **Re: Followup Comments**
> > > > >
> > > > > Thank you for your continued responses; they clarified most of my questions.
> > > > >
> > > > > I just want to follow up on point (1) to make sure I understand what could be causing the source of the (apparently quite large?) performance improvement of CDT over DT in the x-velocity experiments (Table 4). My confusion stems from what could be causing CDT to generalize. If DT takes in an unseen conditioning scalar (eg, $\text{xvel} = 1$) that is between two known conditioning values from training (eg, $\text{xvel} = 0$ and $\text{xvel}=2$), then perhaps you might expect some kind of interpolation to occur. But if an unknown _categorical_ input is given, does that mean the model is being conditioned on a randomly-initialized embedding because it was not seen during training?
> > > > >
> > > > > The numbers in Table 4 are Wasserstein distances, right? Since each x-velocity corresponds to a reward function, as described in Appendix E.5:
> > > > > > we modified the reward function for the cheetah to run with specified velocity (such as ```-np.abs(x_vel - target_vel))```...
> > > > >
> > > > > do you also have the cumulative return for CDT versus DT for comparison? These might be slightly more interpretable.

---

> > > > > > ### Author Response · Authors · 2021-11-26
> > > > > > **Details of Cheetah-Velocity Experiment**
> > > > > >
> > > > > > We have been glad to address your concerns. Let us describe the details of the Cheetah-Velocity experiment in Table 4 and Figure 2 (b).
> > > > > >
> > > > > > We guess CDT is also expected to occur some kind of interpolation. Although we specified target velocities [0.0, 0.1, 0.2, ..., 2.9, 3.0] to gather the experience for the dataset, the collected dataset also contains small amount of other x-velocities. For example, to reach x-velocity to 2.5, the agent definitely passes through 0.0 - 2.4 at the beginning of the episode. We have assumed that the vector histograms may hold and leverage such "small" information, while the scalar summations might ignore it. In addition, since we employ a histogram over the future, not the one-hot vector at the current timestep, it may avoid the randomly-initialized embedding.
> > > > > >
> > > > > > > The numbers in Table 4 are Wasserstein distances, right?
> > > > > >
> > > > > > Yes. Same as the other tables (e.g. Table 2, 3, 5), we reported the Wasserstein metrics. We also provide the cumulative rewards of CDT and DT, where CDT outperforms DT. As we can see in Figure 2 (b), DT tends to cover the dataset supports rather than to fit the specific velocities. We think these trends of DT led to the lower distribution matching performance in Table 4 and the lower cumulative reward as well.
> > > > > >
> > > > > > | | x_vel: 0.5 | x_vel: 1.5 | x_vel: 2.5 |
> > > > > > | ---- | ---- | ---- | ---- |
> > > > > > |**CDT** | -12.107 | -49.320 | -61.068 |
> > > > > > | **DT** | -237.409 | -121.449 | -190.806|

---

> > > > > > > ### Comment · Reviewer_cUy5 · 2021-11-29
> > > > > > > **Re: Cheetah velocity experiment**
> > > > > > >
> > > > > > > Thanks for the clarifications. I have edited my original review and score. (I am just leaving this note in case the review edit does not trigger an email notification.)

---

### Author Response · Authors · 2021-11-17
**Summary of Author Response to All the Reviewers (1/2)**

We would like to appreciate all the reviewers for their insightful comments. We revised the manuscript based on the constructive feedback and suggestions from the reviewers. Our key responses are summarized as below:

**> Title**

We slightly changed the title to avoid confusion with distributional RL works:

**Before:** Distributional Decision Transformer for Hindsight Information Matching

**After: Generalized** Decision Transformer for **Offline** Hindsight Information Matching


**> Evaluation**

We resolved the issue of small order in the evaluation metric (we thank the reviewer 5xwG for raising this) by fixing the bugs contained in the evaluation code and leveraging reliable python libraries to compute Wasserstein-1 distance: https://github.com/pkomiske/Wasserstein.


**> Baseline Method for Comparison**

As the reviewer 5xwG and Ffah suggested, we included the additional baseline algorithms; FOCAL, a metric-based offline meta RL method, and Meta-BC, because our problem settings (offline multi-task SMM/IL) resemble meta learning (see Appendix D for the details of those connections). Throughout the experiments, these methods provided good baseline performances to compare.


**> Additional Experiments**

All of the reviewers have raised their concerns about the diversity of the dataset (medium-expert in D4RL), and 1D reward/x-velocity evaluation.
To deal with them, we added the additional experiments: (1) 2D state-feature distribution matching (xy-velocities of Ant, Appendix E.3, Table 11) to show the scalability to multi-dimensional settings, (2) z-velocity distribution matching with synthesized bi-modal target distribution (HalfCheetah, Section 6.2.1) to show the scalability to other state-feature space, and unseen target distribution under the diverse behavior datasets, and (3) unseen x-velocity distribution matching from meta learning (HalfCheetah running with specified x-velocity, Section 6.2.2) to show the generalization to unknown distributions in the evaluation. Throughout the experiments, our Categorical DT outperforms other baselines or is competitive to Meta-BC baselines.

In addition, Figure 2 provides the qualitative comparison against DT, which cannot handle bi-modal or unseen target trajectories, while CDT can. The video of learned bi-modal behavior is available at https://sites.google.com/view/generalizeddt.


**> Clarification of Unsupervised Experiments**

In the revised paper, we introduced Bi-directional Decision Transformer (BDT), which does not receive any explicit feature function $\Phi$ and rather obtain it through a parameterized aggregator, utilizing a second anti-causal transformer that takes a reverse-order state sequence as an input, and also renamed previous “Unsupervised DT“ as DT-AE,  DT-CPC, or DT-E2E (see Section 5.3). Through the experiment, we found that just adding auto-encoder (AE) or contrastive (CPC) regularizer isn’t sufficient enough to learn $\Phi$ without manual feature specification, and that the aggregator choice (in Figure 1) plays an important role; The results in Table 6 (Section 6.3) show that BDT outperforms all of other learned $\Phi$ variants or Meta-BC and is comparable to CDT or DT (in Table 2) with longer input (N= 50).

**> Connection among the Proposed Methods**

We modified Figure 1 to clarify the connection among our proposed methods. We demonstrated how Generalized Decision Transformer leads to different classes of algorithms **with only small architectural changes**.  For example, DT for offline RL can be recovered with $\Phi(s,a)=r(s,a)$ and $\gamma$-discounted summation as an anti-causal aggregator. We can get Categorical DT (CDT) for offline multi-task state-marginal matching (SMM) when the aggregator is binning, and also get Bi-directional DT (BDT) for offline multi-task imitation learning if the aggregator is a second transformer. The choices of $\Phi(s, a)$ and the aggregator together decide $I^{\Phi}(\tau)$ in the Hindsight Information Matching (HIM) objective discussed in Section 4 and Table 1, where conversely GDT can essentially solve any HIM problem with proper choices of $\Phi$ and aggregator.

---

> ### Author Response · Authors · 2021-11-17
> **Summary of Author Response to All the Reviewers (2/2)**
>
> **> Insight from the Unification as a Hindsight Information Matching Problem**
>
> Our unification in Table 1 not only clarifies the difference or relationship among HIM algorithms, but also provides following novel insights:
>
> - New HIM algorithms can be proposed by simply changing $I^{\Phi}(\tau)$, as we did to propose Categorical DT for distribution-based problems.
> - Given a choice of $I^{\Phi}(\tau)$, new HIM algorithms can be proposed by changing implementation details, such as using “RL" or “BC" as algorithm type, doing “Online” or “Offline” training, and network architectures.  All “Offline” “BC” methods could be adopted easily to “Online” learning through iterative data collections.
> - Only (1) goal-based and (2) multi-task problem can use “RL” as Algorithm Type, while all four, plus our (5) distribution-based, can use “BC”, because “RL” requires optimizing Eq. 4 with respect to the policy, which gets non-trivial for some choices of $I^{\Phi}(\tau)$; e.g. (3)-(5) return-based, full trajectory imitation, or distribution-based problem. “BC” bypasses the need to solve Eq. 4 and therefore is universally applicable to any $I^{\Phi}(\tau)$ or HIM algorithm.
>
>
> **Summary**
>
> We thank all the reviewers again for the detailed and constructive review. It seems that all the reviewers have agreed with our contribution of hindsight information matching formulation and Generalized Decision Transformer framework, and that most of the concerns are raised to the experimental evaluations. We hope our additional tasks with diverse datasets and different state-feature space (2D xy-velocities, or z-velocity) & baseline methods (Meta-BC, and FOCAL) in this revision could address all of your concerns. Let us know any remaining questions or concerns if you have.

---

### Decision · Program_Chairs · 2022-01-20

**Decision:**

Accept (Spotlight)

**Comment:**

The paper describes a framework that unifies several previous lines under hindsight information matching.  Within that framework, the paper also describes variants of the decision transformer (DT) called categorical DT and unsupervised DT.  The rebuttal was quite effective and the reviewers confirmed that their concerns are addressed.  The revised version of the paper is significantly improved and consists of an important contribution that should interested many researchers.  Well done!